



# System design and scaling trends for airborne wind energy

Rishikesh Joshi, Dominic von Terzi, and Roland Schmehl

Delft University of Technology, Faculty of Aerospace Engineering, Kluyverweg 1, 2629 HS Delft, The Netherlands

**Correspondence:** Rishikesh Joshi (r.joshi@tudelft.nl)

**Abstract.**

So far, the size of horizontal axis wind turbines (HAWTs) has steadily increased, but recent studies and market decisions suggest that this trend may come to an end. Airborne wind energy (AWE) is an innovative technology that differs from the operating principles of HAWTs. It uses tethered flying devices, denoted as kites, to harvest higher-altitude wind resources. Kites eliminate the need for a tower but introduce a penalty in power generation since the kite has to spend part of its aerodynamic force to counter its weight. The differences between the two technologies lead to different scaling behaviours, and understanding these and the design drivers of AWE systems is essential for developing this technology further. To this end, we developed a multi-disciplinary design, analysis and optimisation (MDAO) framework which employs models evaluating the wind resource, power curve, energy production, overall component and operation costs, and various economic metrics. This framework was used to design fixed-wing ground-generation (GG) AWE systems based on the objective of minimising the levelised cost of energy (LCoE). The variables used to define the system were the wing area, aspect ratio, tether diameter and rated power of the generator. The framework was employed to find optimal system designs for rated power ranging from 100 kW to 2000 kW. The results show that kite mass, energy storage, and tether replacements are the key LCoE-driving factors. Moreover, in contradistinction to HAWTs, the total lifetime operational costs are equal to or higher than the initial investment costs. This distribution of costs over the project's lifetime, rather than as a large upfront investment, could make it easier to secure project financing. The scaling results show that the LCoE-driven optimum lies within the 100 kW to 1000 kW system size. The reason for this is that the kite mass penalty increases the cut-in and rated wind speeds, reducing the capacity factor of the larger systems. Sensitivity analyses with respect to extreme scenarios considering technological advancements, financial uncertainties and environmental conditions show that this optimum is robust within our modelling assumptions.

## 1 Introduction

Significant progress has been made in the development and scaling of horizontal axis wind turbines (HAWTs) over the past half-century. However, with the increasing sizes, these machines are now encountering challenges in further upscaling due to structural, logistical, and economic constraints (Canet et al., 2021; Mehta et al., 2024a). Airborne wind energy (AWE) is an emerging technology that differs in operating principles from HAWTs. The primary motivation behind developing AWE technology is the hypothesis that for a given location, a similar amount of energy can be produced at a lower cost and a lower carbon footprint compared to a wind turbine. This is because AWE systems access a higher altitude wind resource than wind





turbines (Bechtle et al., 2019) and use less material (Hagen et al., 2023; Coutinho, 2014). Multiple AWE concepts exist and can be classified in various ways. A review of all existing technologies can be found in Vermillion et al. (2021); Fagiano et al. (2022). One classification criterion is the type of flight operation, which can be crosswind or tether-aligned. Another criterion

is the power generation method, which can be fly-generation (FG) or ground-generation (GG). In the FG concept, power is produced onboard using small ram-air turbines and transmitted to the ground via conducting tethers. In the GG concept, the kite pulls the tether, which unwinds a drum-generator module on the ground, generating power. Another GG concept, the rotary system, involves transmitting the torque generated by a network of wings to a ground-based generator via a network of tethers. Additionally, AWE systems can also be classified based on the type of flying device, which includes multiple concepts such as

soft-wing, fixed-wing, and hybrid-wing configurations. Figure 1 shows the analogy between the components of a HAWT and a GG AWE system. The rotor nacelle assembly of wind turbines is structurally held at the designed hub height with the help

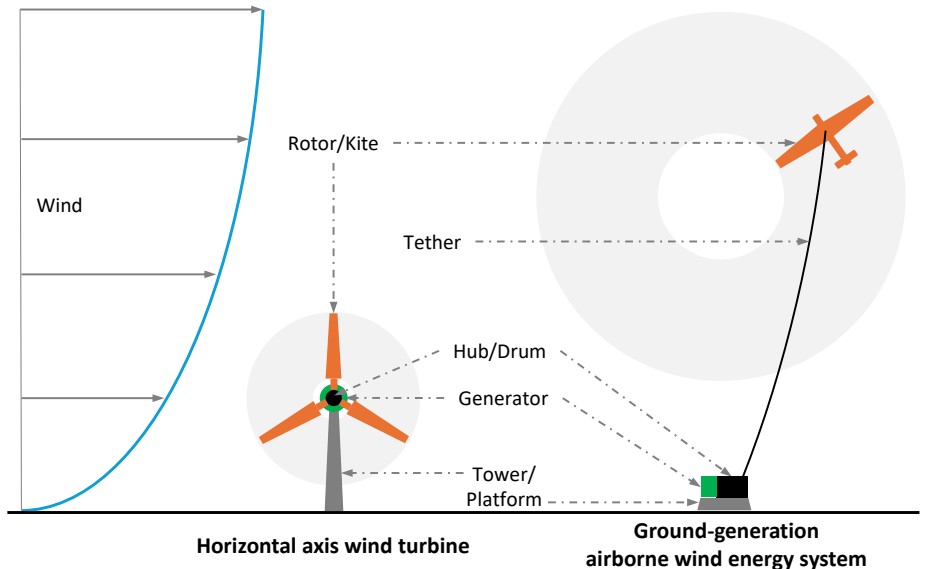

**Figure 1.** Analogy between the components of a horizontal axis wind turbine and a ground-generation airborne wind energy system.

of a tower. In contrast, the kite maintains the required height by spending part of its aerodynamic force to compensate for the gravitational force. A hub is responsible for the torque transfer from wind turbine blades to the generator. While, the GG AWE systems extract power from the thrust (pulling force) generated by the kite, which is then transferred with the help of a tether

to a drum on the ground. The drum is then responsible for converting this linear pulling force into torque, which drives the generator. These differences in components and operational principles suggest that the scaling trends of AWE systems might be different than those of HAWTs.

The present work aims to develop an understanding of the key design drivers, trade-offs and scaling potential of AWE systems. Design drivers are the aspects or parameters that greatly affect performance. Understanding these can lead to a better

understanding of the potential of the technology and its value proposition. This is achieved by employing systems engineering





principles to design AWE systems. The system design of AWE entails the design of the kite, the tether, and the ground station. The ground station consists of the drum, which supports the loads during operation and stores the tether; the drivetrain, which transfers and converts the mechanical power to electrical power; the yawing mechanism, which enables the kite to align with the wind direction; the launch and land system; and the control station.

The power output of GG AWE systems is oscillating by nature and needs to be smoothened to comply with grid codes before the systems can be connected to the electricity grid. This power smoothing can be achieved with intermediate storage components that act as buffers to charge and discharge during the operation (Joshi et al., 2022). Another approach is farm-level power smoothing by operating multiple AWE systems in a phase-shifted but synchronised manner. Although this approach is expected to reduce the requirement of the intermediate storage solution (Faggiani and Schmehl, 2018), it will pose a challenging

active control problem. Three different types of drivetrain configurations, electrical, hydraulic and mechanical, depending on different storage solutions, were explored by Joshi et al. (2022). Because of the commercial readiness and proven track record of comprising components, the electrical drivetrain is considered a more suitable drivetrain for market entry of AWE systems. Therefore, we limit the scope of the present work to the electrical drivetrain shown in Fig. 2. It consists of a gearbox, generator, power electronics and an electrical storage unit, which could be a battery pack or a supercapacitor bank.

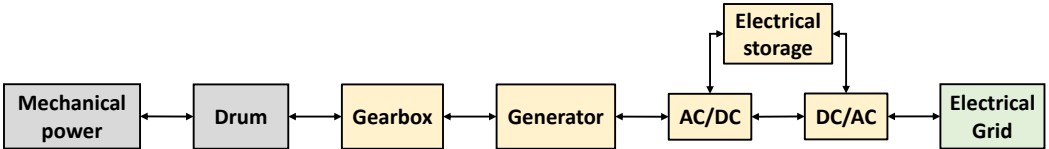

**Figure 2.** Electrical drivetrain architecture (adapted from Joshi et al. (2022)).

Compared to wind turbines, AWE is still in its early development phase, with the first commercial prototypes in the range of up to several hundred kilowatts. Therefore, highly matured and sophisticated models do not yet exist (US Department of Energy (DOE), 2021; Weber et al., 2021). Fast and scalable models that capture all relevant physics are essential for system design studies. Sommerfeld et al. (2022); Joshi et al. (2024); Trevisi and Croce (2024) performed system design and scaling studies focusing on the performance of systems in terms of power or energy production, but in most markets, performance is

measured using a metric known as the levelised cost of energy (LCoE). It is the ratio of the system's total costs to the total energy it can produce over its lifetime. A cost model, along with power and energy production models, is needed to evaluate this metric. Grete (2014); Faggiani and Schmehl (2018) performed LCoE-driven design studies using a quasi-steady model for soft-wing GG system developed by Schmehl et al. (2013); Van der Vlugt et al. (2019) and building upon the cost model proposed by Heilmann and Houle (2013). The kite mass was considered dependent on wing area, but in reality, it is also

dependent on the aspect ratio and the wing loading. The generator-rated power was considered a constant parameter in their optimisation but should be considered as a design variable since it has a significant cost share. Tether lifetime and intermediate storage costs for power smoothing were not modelled. Joshi et al. (2023) introduced a methodology for system design of AWE systems based on cost as well as profit-based metrics, using closed-source company data, which is not accessible in the public





domain. Table 1 lists the LCoE values reported in articles and technical reports available in the public domain. The range of
values is indicative of the uncertainty and diversity in modelling assumptions of different studies. A comprehensive system
design analysis by coupling power, energy production and cost models is still missing in the literature.

**Table 1.** Overview of LCoE values reported in the public domain.

| Source | Concept | Power (kW) | LCoE (€/MWh) |
|---|---|---|---|
| Heilmann and Houle (2013) | Soft-wing GG | [600, 900, 1400]* | [50, 45, 48] |
| Grete (2014) | Soft-wing GG | [600, 900, 1400]* | [43, 50, 60] |
| Gambier et al. (2014) | Soft-wing GG | [20, 200, 2000] | [19-32, 18-22, 25-43] |
| European Commission (2018) | Agnostic | Agnostic | 46-150 |
| Gambier et al. (2017) | Soft-wing GG | [20, 200, 2000] | [51-57, 45-50, 46-51] |
| Faggiani and Schmehl (2018) | Soft-wing GG | 100 | 120 |
| Garcia-Sanz (2020) | Agnostic | 4100 | 72 |
| BVG Associates (2022) | Agnostic | Agnostic | 40-80 |

* maximum reel-out power

Because of the lower maturity level and lack of validation against experiments or measurements, current AWE models exhibit
higher uncertainties. Therefore, the goal is not to estimate an optimal system design with an accurate prediction of LCoE but
to identify the trends and trade-offs within the design space. This work falls within the conceptual design phase. A holistic
and integrated framework is essential to evaluate the impact of a change in a design parameter on the system's performance
with respect to a chosen objective. Such an integrated framework can be used for techno-economic analyses, sizing, scaling,
and system design optimisation studies. The present work is focused on the fixed-wing ground-generation concept, but the
proposed methodology can be applied to any AWE concept depending on the availability of individual models tailored to the
particular concept.

The paper is structured as follows. Section 2 presents the methodology and the underlying models used to identify the design
drivers and trade-offs, Sect. 3 presents a reference case study exploring the system design for rated power ranging from 100
kW to 2000 kW. This rated-power range is chosen since most AWE companies are targeting their first commercial system
within a scale of 100 kW, and some of them are aiming for further upscaling up to a multi-megawatt scale (Sánchez-Arriaga
et al., 2024). This analysis is followed by a sensitivity analysis to capture extreme scenarios compared to the reference case
study, and finally, Sect. 4 presents the key conclusions.

## 2 Methodology

Valuable insights can be gained by examining how the design objective changes with variations in system design variables.
This approach forms the basis for design space exploration and optimisation. By observing the sensitivity to design variables,
we produce an optimal system design as a reference. The developed framework is based on the engineering field of multi-





disciplinary design, analysis, and optimisation (MDAO). MDAO tools and methodologies consider the interactions between different subsystems and disciplines, enabling a comprehensive assessment of a system's performance based on chosen objectives. This approach is often used in aerospace, automotive, and other industries where system complexity and the interplay between components are significant. Section 2.1 discusses the problem formulation and Sect. 2.2 presents the employed system design framework.

## 2.1 Problem formulation

Problem formulation is the description of the design objective, variables and constraints. It depends on the context that defines the settings in which the system will operate. The system's rated power is a requirement set before designing a system. The design objective is usually determined by the requirements set by the market for which the system is designed. This specific market deployment scenario can be defined by the wind conditions, discount rate and price. For a given design objective, certain design variables or constraints can have a high or low influence on the performance of systems. Variables to which the objective is highly sensitive are considered design drivers. Constraints could act as design limiters if they restrict the optimum design. The market and project-specific aspects primarily influence the constraints.

### 2.1.1 Design objective

The design objective is the goal that the developer wants to achieve with the system. The most common but also conflicting objectives are higher energy production and minimum costs. Levelised cost of energy (LCoE) is a holistic metric that combines both objectives into one metric and is defined as

$$
\text{LCoE} = \frac{\sum_{y=0}^{N_y} \frac{\text{CapEx}_y + \text{OpEx}_y}{(1+r)^y}}{\sum_{y=0}^{N_y} \frac{\text{AEP}_y}{(1+r)^y}}, \tag{1}
$$

where $\text{CapEx}$ is the capital expenditure, $\text{OpEx}$ is the operational expenditure, $r$ is the discount rate, AEP is the annual energy produced, $y$ is the instantaneous year, and $N_y$ is the project lifetime.

### 2.1.2 Design variables

Design variables are the system design parameters which the developer can vary to maximise the performance of systems. In the case of wind turbines, rotor size is the variable that limits the power extraction for a given generator size. In contrast, for AWE systems, the combination of the kite and the tether dimensions limits the power extraction. Therefore, for AWE systems, the tether is an additional component that must be designed in coherency with the kite and the generator. Table 2 lists the chosen system design variables, enabling the evaluation of relevant trade-offs with respect to the chosen design objective.

The kite is characterised by the wing area and the aspect ratio. The wing span is a dependent variable. On one hand, with increasing wing area, the aerodynamic force increases, but on the other, the kite mass also increases. This increases the aerodynamic force component to compensate for the gravitational force, reducing the extractable power (Schmehl et al., 2013; Van der Vlugt et al., 2019; Joshi et al., 2024). Moreover, larger wing areas and mass also lead to higher costs due to higher



**Table 2.** Chosen system design variables characterising the kite, the tether and the drivetrain.

| Description | Parameter | Unit |
|---|---|---|
| Wing area | $S$ | $\mathrm{m}^2$ |
| Aspect ratio | AR | - |
| Maximum wing loading | $W_{\mathrm{l,max}}$ | $\mathrm{kNm}^{-2}$ |
| Maximum tether stress | $\sigma_{\mathrm{t,max}}$ | GPa |
| Power crest factor | $f_{\mathrm{crest}}$ | - |

material usage. Higher aspect ratios reduce the induced aerodynamic drag, thereby increasing the aerodynamic efficiency and affecting the kite mass. These trade-offs are critical to capture in the system design process.

The tether is characterised by the maximum allowable wing loading and the maximum allowable tether stress. These limit the maximum force it can handle for a given wing area or the maximum stress with respect to its cross-sectional area, respectively. For a given tether material strength, the larger the tether diameter, the higher the wing loading capacity and the lower the stress in the tether. Higher wing loading capacity enables higher power extraction but also increases drag losses due to the increased tether diameter. Moreover, the kite has to be structurally capable of withstanding higher loading, which leads to a higher kite mass. Higher maximum allowable tether stress reduces the tether diameter but negatively affects the fatigue lifetime of the tether, significantly increasing the replacement costs. These trade-offs are also critical to capture in the design process.

The power crest factor is the ratio of the generator's rated power to the system's rated power. This is relevant because the instantaneous power during a cycle is higher than the cycle average power of AWE systems. This effect is more pronounced for GG systems with reel-out and reel-in phases. Therefore, the drivetrain must be designed according to the peak power during the cycle. The power crest factor indicates the trade-off between capping the power at lower values, which will reduce the net cycle power but will also reduce the overall drivetrain costs.

### 2.1.3 Design constraints

Design constraints are external factors that developers cannot control but must be considered and integrated into the design process. These are primarily derived from project-specific requirements, including safety and regulation requirements (Salma and Schmehl, 2023). Table 3 lists the chosen design constraints.

**Table 3.** Chosen system design constraints to incorporate the project-specific safety and regulation requirements in the design process.

| Description | Constraint | Unit |
|---|---|---|
| Available land area | $A_{\mathrm{oper}} \leq A_{\mathrm{land}}$ | $\mathrm{m}^2$ |
| Operation height limits | $h_{\mathrm{min}} \leq z_{\mathrm{k}} \leq h_{\mathrm{max}}$ | m |





The available land area $A_{\mathrm{land}}$ is usually a project-specific constraint, while the operation height limits $h_{\min}$ and $h_{\max}$ could be driven by safety and regulation requirements. The area of operation $A_{\mathrm{oper}}$ is the ground area density of the system

determined by calculating the circular area using the projected tether length on the ground as the radius. In addition, noise and visual constraints could also be applied, but there is currently insufficient information available to quantify these constraints for AWE systems.

## 2.2   System design framework

An integrated system design framework based on MDAO methodology is developed, incorporating models that cover wind re-

sources, power production, energy production, and economics. Lambe and Martins (2012) described a methodology to present the MDAO frameworks in a formalised manner through an extended design structure matrix (XDSM). Figure 3 shows the XDSM of the developed MDAO framework.

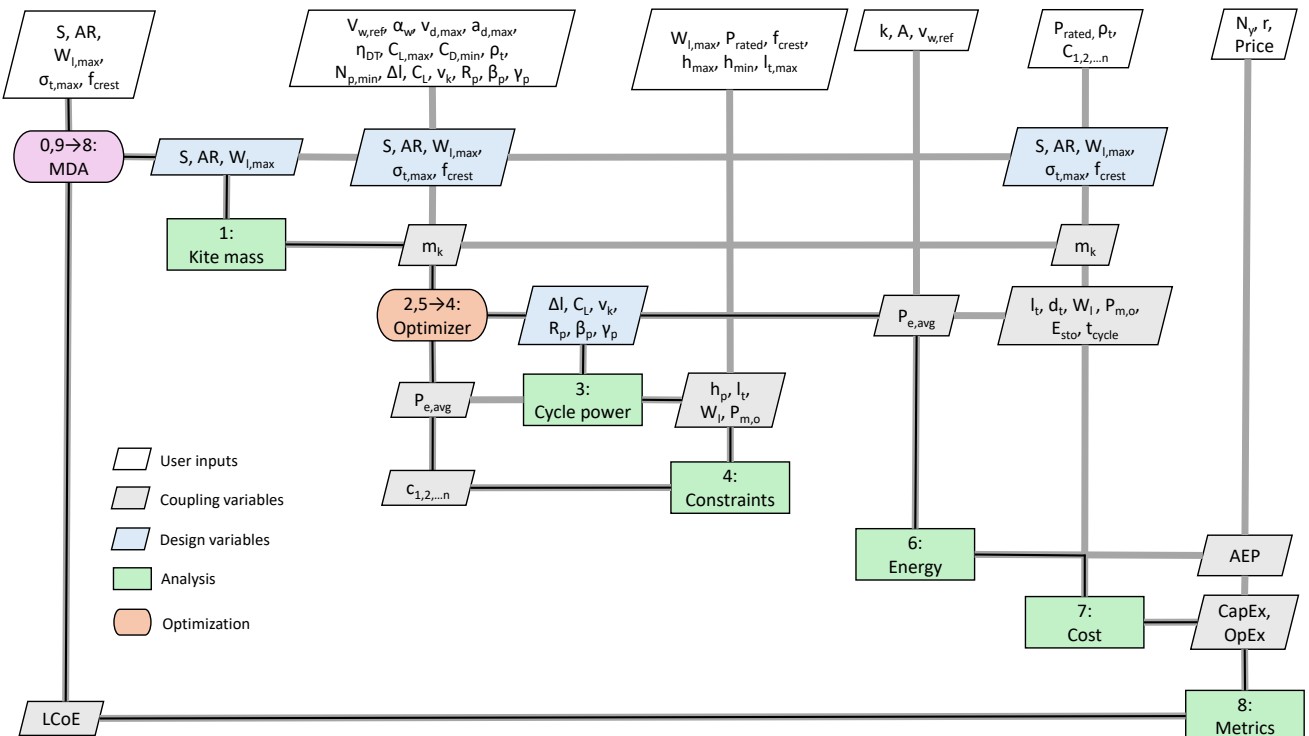

**Figure 3.** Extended design structure matrix (XDSM) of the developed design framework for airborne wind energy systems. The matrix illustrates the framework's workflow, with the MDA block controlling the process. Thick grey lines represent data flow, while thin black lines indicate process flow. Vertical connections show inputs, and horizontal connections show outputs. Blue blocks denote design variables, grey blocks represent coupling variables, and white blocks indicate user inputs defined at the start. Loops are marked by $m \rightarrow n$, where $m \leq n$, and sequential numbering defines block execution order.




In the diagram, various block shapes and colours denote inputs, outputs, computational processes and loops. The MDA block is the controlling block that defines the framework's employed workflow. Thick grey lines denote the flow of data within the framework. Vertical connections from top to bottom denote input to the subsequent blocks, whereas the horizontal connections on either side of the blocks denote the outputs from the particular blocks. The execution order of blocks is denoted by sequential numbering starting from zero. If a block is a start and an endpoint for a loop, it is denoted by two numbers denoting the start and the end, respectively. A loop is denoted by $m \rightarrow n$ where $m \leq n$. The thin black line within the thicker grey lines denotes the process flow. The user inputs are defined in the white blocks at the top, which need to be defined initially; the design variables are denoted by the blue blocks, and the coupling variables between computation blocks are denoted in grey blocks.

One iteration of the framework starts and ends at the MDA block denoted by '0, 9' and includes the evaluation of all process blocks from 1 to 8. This evaluates the LCoE value for the input design based on the initialisation of variables and constraints. This workflow can be deployed to evaluate a single design, a design space with many combinations of variables, or as an optimisation problem in which the system design variables are optimised until a chosen convergence criteria (e.g. minimum LCoE) is satisfied.

The following sections describe the models used in the framework.

## 2.3 Kite mass model

The kite mass model used in the present work was introduced in Joshi et al. (2024). It is a non-linear data-driven model based on the Ampyx Power 20 and 150 kW prototypes and their MW scale design projections. The parametric dependency can be represented as

$$m_{\mathrm{k}} = f(S, AR, W_{\mathrm{l,max}}). \tag{2}$$

The kite mass increases with increasing wing area due to the increased size of the kite and with increasing aspect ratio since slender wings require more material to maintain stiffness. Higher wing loading demands a structurally stronger kite, which also increases the kite mass. A more refined estimate of the kite mass is a complex interdisciplinary process which requires coupled aero-structural models such as in Eijkelhof and Schmehl (2024). These models require knowledge of many design and manufacturing details that are not known at the conceptual level and are highly specific to the design. They are also computationally demanding. We consider a simple mass model for the present work since we are not focused on absolute values but on the design trends within a large design space. The kite mass model in the present framework could be replaced by a coupled aero-structure model, which can give better mass estimates.

## 2.4 Cycle power model

Also, the cycle power model for pumping cycle systems used in the present work was introduced in Joshi et al. (2024). The conceptual setup of this model is illustrated in Fig. 4. It is a steady-state model implemented within an optimisation algorithm which maximises the mean electrical cycle power $P_{\mathrm{e,avg}}$. This model is analogous to models maximising the coefficient of power ($C_{\mathrm{p}}$) of wind turbines by optimising the operational parameter, the tip speed ratio. In our case, the operational parameters



which are optimised are the reel-out stroke length $\Delta l$, wing lift coefficient $C_{\mathrm{L}}$, kite speed $v_{\mathrm{k}}$, pattern radius $R_{\mathrm{p}}$, pattern elevation angle $\beta_{\mathrm{p}}$ and cone opening angle $\gamma_{\mathrm{p}}$. The stroke length is divided into several segments, and a single flight state is assigned per segment, resulting from the force equilibrium solved for that segment. The orange dots represent these numerical evaluation points, with the dots on the central axis representing the reel-out phase and the upper conical axis representing the reel-in phase. This setup can be used to compute the cycle power for any user-defined wind speed range and a vertical wind speed profile.

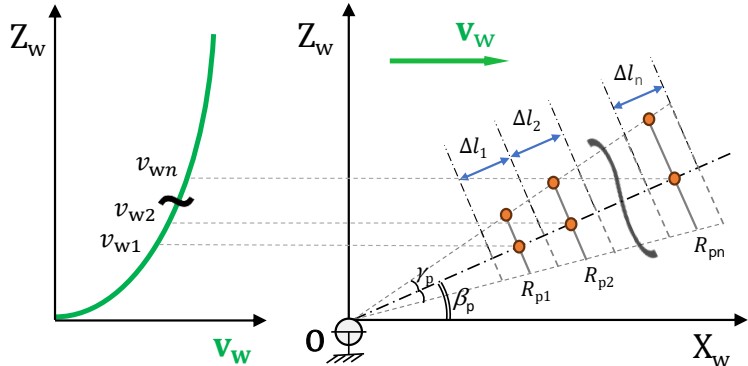

**Figure 4.** Conceptual illustration of the cycle-power model: vertical wind velocity profile (left) and discretized cycle trajectory (right), as described in Joshi et al. (2024).

The cycle power is dependent on the system design variables listed in Table 2 and the output of the kite mass model. This dependency can be represented as

$$P_{\mathrm{e,avg}} = f(v_{\mathrm{w}}, m_{\mathrm{k}}, S, AR, W_{\mathrm{l,max}}, \sigma_{\mathrm{t,max}}, f_{\mathrm{crest}}). \tag{3}$$

The interplay between all the above parameters significantly affects the power output of AWE systems. Since the kite traverses a large volume of space during operation, the spatial variation in wind is more relevant than for wind turbines. Therefore, the vertical wind shear profiles are essential to consider in power computation. The present work uses the characterization of vertical wind profiles in neutral atmospheric conditions using the power law given by (Peterson and Hennessey, 1978).

$$v_{\mathrm{w}}(z) = v_{\mathrm{w,ref}}(h_{\mathrm{ref}}) \left( \frac{z}{h_{\mathrm{ref}}} \right)^{\alpha_{\mathrm{w}}}. \tag{4}$$

This law describes the relationship between wind speed $v_{\mathrm{w}}$, height at evaluation point $z$, reference height $h_{\mathrm{ref}}$, and ground surface roughness parameter $\alpha_{\mathrm{w}}$. Another important input to the model is aerodynamic properties. The present framework does not employ an aerodynamic model; hence, the airfoil properties are assumed constant for all the explored designs. These values are based on higher-fidelity aerodynamic analyses, such as in (Vimalakanthan et al., 2018). In addition to the initial guesses of the operational parameters, other fixed inputs to the model are the volumetric tether material density $\rho_{\mathrm{t}}$, maximum reeling speed $v_{\mathrm{drum,max}}$ and acceleration $a_{\mathrm{drum,max}}$ of the drum, and efficiencies of the drivetrain components $\eta_{\mathrm{DT}}$.



Along with the electrical cycle average power $P_{\mathrm{e,avg}}$, the model output includes the operating height of the kite $h_{\mathrm{p}}$, the deployed tether length $l_{\mathrm{t}}$, wing loading $W_{\mathrm{l}}$, etc. for the entire wind speed range. The optimiser must also respect the constraints on operational and system design parameters as listed in Table 3. The kite drag, tether mass and tether drag contributions are updated in every simulation iteration and depend on operational and system design variables. This also has a significant impact on the performance of the system. The kite drag coefficient has a lift-induced drag component given as

$$C_{\mathrm{D,k}} = C_{\mathrm{d,min}} + \frac{(C_{\mathrm{L}} - C_{\mathrm{l,Cd,min}})^2}{\pi A R e}, \tag{5}$$

where $C_{\mathrm{d,min}}$ is the parasitic drag, $C_{\mathrm{L}}$ is the wing lift coefficient, and $C_{\mathrm{l,Cd,min}}$ is the lift coefficient at $C_{\mathrm{d,min}}$. Anderson (2016) gives the dependency of the wing planform efficiency factor $e$, also known as the Ostwald efficiency, on AR as

$$e = 1.78(1 - 0.045 AR^{0.68}) - 0.64. \tag{6}$$

This relation is based on empirical data from aircraft. A higher operational $C_{\mathrm{L}}$ and lower AR increases the induced drag and vice-versa. The tether mass is a function of the instantaneous tether length and diameter whose dependency can be represented as

$$m_{\mathrm{t}} = f(l_{\mathrm{t}}, d_{\mathrm{t}}), \tag{7}$$

where,

$$d_{\mathrm{t}} = f(W_{\mathrm{l,max}}, \sigma_{\mathrm{t,max}}). \tag{8}$$

Higher $W_{\mathrm{l,max}}$ and lower $\sigma_{\mathrm{t,max}}$ leads to a larger tether diameter and vice-versa. The tether drag is lumped at the kite, and its dependency can be represented as

$$D_{\mathrm{t}} = f(S, d_{\mathrm{t}}, l_{\mathrm{t}}). \tag{9}$$

The larger the wing, the lower the effective contribution of the tether drag, whereas the thicker and the longer the tether, the higher its contribution.

## 2.5 Energy production model

This model is based on the standard approach used in wind energy to compute energy production. The energy produced over a year, also known as annual energy production (AEP), depends on the wind resource at the location and the power curve based on that location's vertical wind shear profile. AEP is calculated as

$$\mathrm{AEP} = 8760 \int\limits_{v_{\mathrm{w,cut-in}}}^{v_{\mathrm{w,cut-out}}} P_{\mathrm{e,avg}}(v_{\mathrm{w,ref}}) f(v_{\mathrm{w,ref}}) \mathrm{d}v_{\mathrm{w,ref}}, \tag{10}$$

where $v_{\mathrm{w,ref}}$ is the wind speed at the chosen reference height $h_{\mathrm{ref}}$, $v_{\mathrm{w,cut-in}}$ and $v_{\mathrm{w,cut-out}}$ are the cut-in and cut-out wind speeds, respectively, and $f(v_{\mathrm{w,ref}})$ is the probability of occurrence of wind speeds in a year. It is assumed that the wind





characteristics remain constant over each year, such that the energy produced over the entire lifetime is calculated as

$$E_{\text{lifetime}} = \sum_{y=0}^{N_{\text{y}}} \frac{\text{AEP}_{\text{y}}}{(1+r)^y},$$ (11)

where $r$ is the discount rate, $y$ is the instantaneous year, and $N_{\text{y}}$ is the project lifetime in number of years.

### 2.6 Cost model

The cost model used in the present work is described in Joshi and Trevisi (2024). This model was developed as a collaborative effort between industry and academia as a part of the IEA Wind TCP Task 48 (IEA Wind TCP, 2021). It includes parametric cost models for both capital expenditure (CapEx) and operational expenditure (OpEx) associated with each subsystem of an AWE system: the kite, the tether and the ground station. A workshop was conducted as part of this task to collect inputs and reference data. Ten participants from this workshop provided significant input in building this model. The portfolio of participants who provided input includes AWE companies, tether and ground station manufacturers, suppliers, and university research groups. In addition to the input from participants, publicly available reports and articles (Heilmann and Houle, 2013; Grete, 2014; BVG Associates, 2019; Stehly et al., 2020; Ramasamy et al., 2022; BVG Associates, 2022) were also used to collect cost references. The cost model is thus a combination of industry data for AWE-specific components such as the kite and the tether, off-the-shelf price data for generic components, and physics-based estimations for lifetime estimations.

By their nature, cost models are highly uncertain because they are subject to nonscientific, nontechnical, site-dependent, and sometimes political considerations. Therefore, many assumptions must be considered in the derivation, especially at the current early stage of technology development. The cost references provided in this report are based on the early commercialization of AWE systems with the system sizes ranging from 100 kW - 2000 kW and series production volumes of 50+ units. Moreover, we do not consider any overhead costs in development, manufacturing and profit margins, which might be significant for certain low TRL (technology readiness level) and CRL (commercial readiness level) components. The following sections detail the modelling of the subsystem cost contributions.

#### 2.6.1 Kite costs

The kite costs are divided into structure costs $C_{\text{k,str}}$ and avionics costs $C_{\text{k,avio}}$. The kite structure costs are dependent on the kite mass and wing area. The dependence on the mass is due to the cost of the composite material, adhesive, and production. The dependence on the wing area is due to the costs of surface treatment, coating, etc. In this case, the material is a carbon-fibre composite and the costs are modelled as

$$C_{\text{k,str}} = p_{\text{str}} m_{\text{k}} + p_{\text{S}} S,$$ (12)

where $m_{\text{k}}$ is the kite mass from Eq. 2 and $S$ is the wing area. For a better estimate, this cost function should include the total surface area of the kite, which includes the wing, fuselage, tail surfaces, and any other parts that are exposed to airflow during flight. $p_{\text{str}}$ is 250 €/kg and $p_{\text{S}}$ is 200 €/m². Structural costs are expected to drop with the development of technology and with improved structural designs, but such future projections are not considered in the present work.





The avionics costs do not scale with the size of the kite and, hence, can be considered fixed. They typically include all the electronic systems used on the kite, such as communication and navigation hardware, sensors, CPUs, and any electronic system needed to perform individual functions. Usually, these have a high share in costs due to the requirements for aviation-grade certification and redundancies. For prototypes, the avionics cost is estimated to be $C_{\mathrm{k,avio}}$ = 15 k€. The aviation certification and redundancy requirements are expected to raise the $C_{\mathrm{k,avio}}$ = 150 k€ for early commercial production.

The kite will have other cost components, such as the tether attachment mechanism and protection equipment necessary in extreme events or to ensure longer life, such as lightning protection, de-icing, erosion protection, etc. It will also have some other maintenance costs over the lifetime, but these are not modelled due to lack of information.

### 2.6.2 Tether costs

The tether is a structural component which has to withstand the pulling force of the kite. The price in €/kg depends on the type of material and the suppliers. The tether fibre commonly used in the AWE industry is the Dyneema fibre (Bosman et al., 2013). Multiple strands are usually braided together to manufacture the tether, and hence, the tethers have a hollow inner core. The nominal diameter is the measured diameter of a newly manufactured tether. After experiencing some loading, this diameter becomes smaller and is called the worked-in diameter. This is supposed to be used in performance evaluations. A hollow core of 15% of the cross-sectional area can be assumed, such that the stress acting on the tether

$$\sigma_{\mathrm{t}} = \frac{F_{\mathrm{t}}}{f_{\mathrm{At}} \pi \frac{d_{\mathrm{t}}^2}{4}}, \tag{13}$$

where $F_{\mathrm{t}} = W_{\mathrm{l}}S$ in N, $f_{\mathrm{At}} = 0.85$, which is the ratio between the cross-sectional area taken by the fibres and the tether cross-sectional area, and $d_{\mathrm{t}}$ is the worked-in tether diameter. Different wear-resistant coatings are usually applied on the tether, which increases its total mass. This is usually around 10% of the total mass. The tether CapEx can be computed as

$$C_{\mathrm{t}} = p_{\mathrm{t}} m_{\mathrm{t}}, \tag{14}$$

where $p_{\mathrm{t}}$ is 80 €/kg and $m_{\mathrm{t}}$ is determined from Eq. 7.

Bosman et al. (2013) described the design drivers for the tether and highlighted that bending fatigue and creep as the leading causes of tether failure. The bending fatigue is mainly relevant for GG systems since it arises when the tether is unwound from the drum at high tension. The bending failure is estimated using the method described in Bosman et al. (2013). The number of cycles to failure $N_{\mathrm{b}}$ is a function of the ratio between the drum diameter $d_{\mathrm{drum}}$ and the tether diameter $d_{\mathrm{t}}$, and the tether stress. It is given as

$$N_{\mathrm{b}} = 10^{(a_1 - a_2 \sigma_{\mathrm{t}})}, \quad \text{for} \quad 0.2 < \sigma_{\mathrm{t}} < 0.8 \, \mathrm{GPa}, \tag{15}$$

where $\sigma_{\mathrm{t}}$ is in GPa and the values of $a_1$ and $a_2$ are dependent on $d_{\mathrm{drum}}/d_{\mathrm{t}}$ and are given in Table 4. The number of cycles to failure with respect to the stress levels for the given $d_{\mathrm{drum}}/d_{\mathrm{t}}$ ratios is shown in Fig. 5.

Miner's rule is commonly used in fatigue life analysis. It is based on the assumption that damage accumulates linearly with each load cycle. This means that the total damage caused by different stress cycles can be summed up to predict when the



**Table 4.** Parameter $a_1$ as a function of the drum to tether diameter ratio $d_{\mathrm{drum}}/d_t$ and constant parameter $a_2$.

| $d_{\mathrm{drum}}/d_t$ | 10 | 20 | 30 | 100 |
|---|---|---|---|---|
| $\mathbf{a_1}$ | | 5.4 | 5.8 | 6.1 | 6.5 |
| $\mathbf{a_2}$ | | | 2.6 | | |

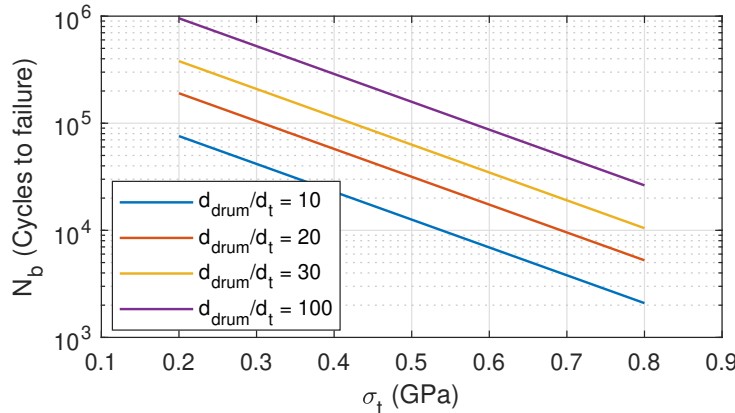

**Figure 5.** Relation between the number of cycles to failure and tether stress for different $d_{\mathrm{drum}}/d_t$ ratios.

material will fail. Using the Miner's rule, for a given wind distribution $f(v_{\mathrm{w,ref}})$, a tether failure will occur when

$$L_{\mathrm{t,bend}}8760N_{\mathrm{bends}}\int_{v_{\mathrm{w,cut-in}}}^{v_{\mathrm{w,cut-out}}}\frac{f(v_{\mathrm{w,ref}})}{t_{\mathrm{cycle}}(v_{\mathrm{w,ref}})N_b(v_{\mathrm{w,ref}})}\mathrm{d}v_{\mathrm{w,ref}}=1, \qquad (16)$$

where $L_{\mathrm{t,bend}}$ is the tether lifetime, $N_{\mathrm{bends}}$ is the number of times the tether bends per cycle. There is at least one pulley in addition to the drum, which guides the tether during winding, and hence, we assume $N_{\mathrm{bends}}=2$. $v_{\mathrm{w,cut-in}}$ and $v_{\mathrm{w,cut-out}}$ are the cut-in and cut-out wind speeds respectively and $t_{\mathrm{cycle}}$ is in h. The frequency of tether replacement per year can, therefore, be calculated as

$$f_{\mathrm{t,repl}}=\frac{1}{L_{\mathrm{t,bend}}}. \qquad (17)$$

Thus, the tether OpEx per year due to replacements is

$$O_{\mathrm{t}}=f_{\mathrm{t,repl}}C_{\mathrm{t}}. \qquad (18)$$

### 2.6.3 Ground station costs

The modelled ground station costs include the drum and the electrical drivetrain costs. In this drivetrain, the generator is directly connected to the drum with or without a gearbox. During cycle operation throughout the windspeed range, the rotational speed



and torque of the drum vary within a wide range. Hence, a gearbox is generally necessary to convert the rotational speed and torque values to the generator's operational range. A gearbox could be avoided if the generator is custom-designed according to the operation of the AWE system. The generator is connected to an electrical storage and the grid via power converters. The storage solution has to be charged and discharged during the cycle to maintain smooth power output at the grid side.

The following sections detail the individual cost models of the ground station components.

**Drum**

The drum converts the tractive power of the kite into shaft power in the drivetrain. Essentially, the drum is a hollow cylinder with a certain wall thickness. The costs for control and winding mechanisms, including pulleys, guide rails, etc, are not considered due to the unavailability of data. These costs will affect the absolute costs but might not scale significantly with size. Therefore,
the cost of the drum is assumed to be proportional to its mass. The drum is typically made of aluminium or steel. Data on these materials are listed in Table 5. The drum mass can be computed using the tether diameter as the rolling pitch (Heilmann and Houle, 2013). When the tether is wound around a drum, it wraps in a helical pattern due to the drum's diameter and the tether's thickness. When the drum rotates once, the tether advances along the drum by one tether diameter, and this distance is known as the rolling pitch. A safety margin of around $10\%$ is generally used on the tether diameter to calculate the pitch. The drum
also has some dead windings that are not used. Hence, a safety factor on the tether length must also be applied. The drum mass is computed as

$$m_{\mathrm{drum}} = \frac{\pi \left[ d_{\mathrm{drum}}^2 - (d_{\mathrm{drum}} - 2t_{\mathrm{drum}})^2 \right]}{4} \frac{l_{\mathrm{t}} f_{\mathrm{s},1}}{\pi d_{\mathrm{drum}}} d_{\mathrm{t}} f_{\mathrm{s},2} \rho_{\mathrm{mat}}, \tag{19}$$

where $d_{\mathrm{drum}}$ is the external diameter of the drum, $t_{\mathrm{drum}}$ its wall thickness, $d_{\mathrm{t}}$ the tether diameter, $f_{\mathrm{s},1}$ is the safety factor on tether diameter, $l_{\mathrm{t}}$ the tether length, $f_{\mathrm{s},2}$ is the safety factor for tether length, and $\rho_{\mathrm{mat}}$ the material density. The first fraction
represents the cross-sectional area, the second fraction represents the number of windings of the tether around the drum, and the third term represents how much axial space is needed for each winding multiplied by the tether material density. Considering the tether lifetime due to bending, the ratio $d_{\mathrm{drum}}/d_{\mathrm{t}}$ is assumed to be 100, and both safety factors are assumed to be 1.1.

**Table 5.** Drum-related data for aluminium and steel materials.

| Parameter | Unit | Aluminium | Steel |
|---|---|---|---|
| $p$ | €/kg | 10 | 7 |
| $\rho$ | kgm$^{-3}$ | 2700 | 7850 |
| $\hat{\sigma}$ | MPa | 300 | 500 |

We propose a simple first-order engineering approach to design the drum thickness. The maximum tether force $F_{\mathrm{t,max}} = \hat{\sigma}_{\mathrm{t}} \pi d_{\mathrm{t}}^2 / 4$, where $\hat{\sigma}_{\mathrm{t}} = 1.5$ GPa is the tether fibre (Dyneema DM20) strength, assuming no hollow core in this section. We
assume that the drum should withstand the same force, distributed over a rectangular area of width $d_{\mathrm{t}}$ and height $t_{\mathrm{drum}}$, i.e.





$F_{t,max} = \hat{\sigma}_{mat} d_t t_{drum}$, where $\hat{\sigma}_{mat}$ is the tensile strength of the drum material. Therefore, the drum thickness can be correlated to the tether diameter as

$$t_{drum} = \frac{\pi \hat{\sigma}_t}{4 \hat{\sigma}_{mat}} d_t. \tag{20}$$

This simplified method neglects the effects such as the force distribution over the entire drum, stress concentrations, and dynamic loading. We used an additional safety factor of 2 on the estimated thickness of the drum to account for the neglected effects. For a steel drum, the CapEx is computed as

$$C_{drum} = p_{st} m_{drum}, \tag{21}$$

where $m_{drum}$ is computed using Eqs. 19 and 20.

**Gearbox**

Since the gearbox connects the drum to the generator, it has to be sized for the peak mechanical loading during the reel-out phase. The cost and size of the gearbox are not only driven by the transferred shaft power but also by the transferred torque. The benefit of using a gearbox is that it reduces generator costs by controlling the input speed and torque of the generator. Scaling the gearbox costs with transferred power and torque together will give better estimates, but we model the costs only with power due to limited data availability. The costs are modelled as

$$C_{gb} = p_{gb} f_{crest} P_{rated}, \tag{22}$$

where $p_{gb}$=70 €/kW.

**Generator**

The cost of the generator depends on the torque as well as on speed. High torque requires more robust components within the generator, whereas high speed requires high precision, wear resistance, etc. Due to the unavailability of detailed data, we represent the costs as a linear function of the rated power, given as

$$C_{gen} = p_{gen} f_{crest} P_{rated}, \tag{23}$$

where $p_{gen} = 120$ €/kW.

**Electrical energy storage**

The objective of energy storage is to act as an intermediate energy exchanger to charge and discharge during cycle operation to maintain the average cycle power at the grid side. The amount of storage required will be driven by the energy exchange required for this purpose (Joshi et al., 2022). Typical implementations of electrical storage technologies are an ultracapacitor bank or a battery bank. Both have different requirements for sizing as well as different cost and lifetime specifications. While





ultracapacitors can withstand high C-rates (charge-discharge rates) of 100C or more, batteries typically have a low C-rate of around 0.5-1C. This drives the sizing of the two options. A 1C rate means the discharge current will discharge the entire battery
in 1 hour.

In the present work, we choose ultracapacitors as the electrical energy storage component. An ultracapacitor bank is a high-capacity energy storage system composed of multiple ultracapacitor modules connected in parallel or series. Unlike traditional batteries, ultracapacitors store energy electrostatically, enabling rapid charge and discharge cycles with high efficiency. They are commonly used in applications requiring burst power delivery, energy recuperation, fast charging capabilities, and tolerance
to frequent cycling. The costs are modelled as

$$C_{\text{uc}} = p_{\text{uc}} E_{\text{rated,uc}}. \tag{24}$$

where $p_{\text{uc}}$ = 60 k€/kWh and $E_{\text{rated,uc}}$ is the required storage sizing in kWh. This is driven by the maximum energy the ultracapacitor bank exchanges during the cycle operations for all wind speeds.

As an example, Fig. 6 shows the simulated instantaneous and average cycle power of a system with an electrical rated power
of 150 kW at its rated wind speed. $P_{\text{m}}$ and $P_{\text{e}}$ are the instantaneous mechanical and electrical power, respectively. $P_{\text{m,avg}}$ and $P_{\text{e,avg}}$ are the cycle-average mechanical and electrical power, respectively. $P_{\text{m,avg}}$ is a hypothetical cycle average power computed by excluding all the drivetrain efficiencies to estimate the intermediate storage requirement.

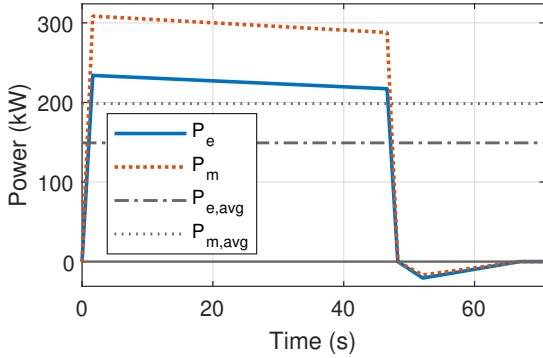

**Figure 6.** Instantaneous mechanical and electrical power over a representative pumping cycle and corresponding average cycle power.

The amount of energy exchanged through the storage during each cycle can be estimated as

$$E_{\text{sto}} = (P_{\text{m}} - P_{\text{m,avg}})t_{\text{o}} = (P_{\text{m,avg}} + P_{\text{m,i,avg}})t_{\text{i}}, \tag{25}$$

where $t_{\text{o}}$ and $t_{\text{i}}$ are the reel-out and reel-in times at respective wind speeds in the entire operation range. The capacity of the ultracapacitor bank is the maximum amount of energy stored in the entire operational range and can be computed as

$$E_{\text{rated,uc}} = \max(E_{\text{sto}}). \tag{26}$$





The ultracapacitors' lifetime depends on the number of charge-discharge cycles for the specific AWE system based on its operational behaviour. The number of charge-discharge cycles in a year is computed as

$$N_{\text{cycles,uc}} = \frac{8760}{E_{\text{rated,uc}}} \int_{v_{\text{w,cut-in}}}^{v_{\text{w,cut-out}}} f(v_{\text{w,ref}}) \frac{E_{\text{sto}}(v_{\text{w,ref}})}{t_{\text{cycle}}(v_{\text{w,ref}})} dv_{\text{w,ref}}. \tag{27}$$

Therefore, the frequency of replacement per year of the ultracapacitor bank is computed as

$$f_{\text{uc,repl}} = \frac{N_{\text{cycles,uc}}}{N_{\text{uc}}}, \tag{28}$$

where $N_{\text{uc}}$ is the indicated lifetime by the manufacturer. It is typically around $10^6$ cycles. Thus the ultracapacitor OpEx per year due to replacements is

$$O_{\text{t}} = f_{\text{uc,repl}} C_{\text{uc}}. \tag{29}$$

**Power converters**

Power converters are electronic devices that convert electrical energy from one form to another, commonly alternating current (AC) to direct current (DC) or vice versa. They regulate the voltage, frequency, and waveform to match the requirements of various electrical systems, facilitating efficient energy transfer. The two power converters in this drivetrain will be sized differently. The converter connected to the generator will be sized according to the generator power rating, whereas the converter on the grid side will be sized according to the rated power of the AWE system. The cost of the power converters is modelled as

$$C_{\text{pc}} = p_{\text{pc}} P_{\text{rated}} (f_{\text{crest}} + 1), \tag{30}$$

where $p_{\text{pc}}$ is 100 €/kW.

The launch and land system (LLS) launches the kite and controls its descent for landing. The LLS will be different for different AWE concepts. There are two commonly used approaches: 1) Horizontal take-off and landing (HTOL), which either uses a catapult or a rotating arm and 2) Vertical take-off and landing (VTOL), which uses electric propellers. On the one hand, HTOL has much larger spatial requirements than VTOL, which significantly drives up the cost of the supporting infrastructure, but on the other hand, VTOL significantly drives up the kite's structural mass and, consequently, the cost. VTOL is most certainly the preferred design choice for FG systems since they already have ram-air turbines which can be used as propellers. Due to the unavailability of data, the LLS costs, along with the yaw system and control station costs, are not modelled in the present work.

### 2.6.4 Balance of system costs

The balance of system (BoS) for a single AWE system is defined as all components except the primary system components, which are the kite, the tether, and the ground station. These costs are more relevant for the evaluations of specific business cases and will be highly dependent on the type and size of the system as well as on site-specific considerations. These considerations





can cause order-of-magnitude changes in the results for different scenarios. The costs considered in the present work are under the assumption of an onshore installation. BoS costs consist of site preparation, foundation, installation, operation maintenance and decommissioning.

Costs under site preparation include removing obstacles such as vegetation, debris, and uneven terrain that could interfere with the kite's launch, flight, or landing. Additionally, any necessary groundwork, such as levelling the surface or installing protective barriers, may be undertaken to optimise the site for efficient and uninterrupted system operation. These costs are modelled as

$$C_{\mathrm{sitePrep}} = p_{\mathrm{sitePrep}} P_{\mathrm{rated}}, \tag{31}$$

where $p_{\mathrm{sitePrep}} = 40\,\text{€}/\mathrm{kW}$.

Foundations and support structures are designed to withstand the forces generated by the AWE system during operation and support the ground station weight. These foundations can vary in design depending on soil conditions, site location, and system requirements. The launch and land apparatus is also an important cost driver for this component. Moreover, these costs will be significantly different for onshore, offshore bottom-fixed and offshore floating scenarios. Since these costs would be driven by the peak power, they are modelled as

$$C_{\mathrm{found}} = p_{\mathrm{found}} f_{\mathrm{crest}} P_{\mathrm{rated}}, \tag{32}$$

where $p_{\mathrm{found}} = 55\,\text{€}/\mathrm{kW}$.

Installation and commissioning involve assembling and configuring components to ensure proper functionality and performance. This process includes erecting support structures, connecting power and communication systems, and testing operational parameters. In addition, commissioning involves fine-tuning control algorithms, conducting safety checks, and verifying compliance with regulatory standards. These costs are modelled as

$$C_{\mathrm{install}} = p_{\mathrm{install}} P_{\mathrm{rated}}, \tag{33}$$

where $p_{\mathrm{install}} = 40\,\text{€}/\mathrm{kW}$.

The operation and maintenance costs include all the yearly costs, for example, the lease of the land used and the insurance costs against potential risks and liabilities associated with their deployment and operation. These costs are modelled as

$$O_{\mathrm{BoS}} = p_{\mathrm{BoS,O}} P_{\mathrm{rated}} \tag{34}$$

where $p_{\mathrm{BoS,O}} = 60\,\text{€}/\mathrm{kW\,year}^{-1}$.

Decommissioning entails safely dismantling and removing components at the end of their operational lifespan or in case of system retirement. This process involves disassembling support structures, disconnecting power and communication systems, and responsibly disposing of materials in accordance with environmental regulations. These costs are modelled as

$$C_{\mathrm{decomm}} = 0.5 C_{\mathrm{install}}, \tag{35}$$





where $C_{\text{install}}$ are the installation and commissioning costs from Eq. 33.

Another type of cost is related to the balance of plant (BoP), which is defined as all components of an AWE farm, excluding the individual system costs. These costs will be relevant for evaluating specific business cases and layout design. BoP includes the array cables, substations and grid integration costs. These costs are not modelled in the present work due to the lack of
information and the focus on system design.

## 2.7    Metrics

As discussed in Sect. 2.1.1, LCoE is the chosen design metric within the present work. This choice of metric is more suitable for scenarios in which the revenue scheme is dependent on subsidies rather than fluctuating electricity market prices. The revenue scheme might shift from subsidy-dependent to market-dependent in future scenarios with high technological maturity.
In such scenarios, profit-based metrics would be more relevant than cost-based metrics such as the LCoE.

Some of these metrics were defined by de Souza Range et al. (2016); Simpson et al. (2020); Joshi et al. (2023); Mehta et al. (2024b). The levelised profit of energy (LPoE) is the difference between the levelised revenue of energy (LRoE) and LCoE (Joshi et al., 2023). The cost of valued energy (CoVE) informs about the ratio of costs to revenue (Simpson et al., 2020). CoVE is similar to LCOE, with the difference that CoVE weighs energy based on the day-ahead market price. CoVE takes
the same value as LCOE when the electricity prices are constant. The net present value (NPV) is the discounted value of the cash flow over the lifetime. The internal rate of return (IRR) is used to estimate the profitability of potential investments. de Souza Range et al. (2016) proposed modified internal rate of return (MIRR) and Mehta et al. (2024b) incorporated it in the design process for HAWTs. MIRR was defined to overcome the limitation of IRR, which is that the positive cash flow from the project is reinvested at the IRR. Comparing the changes in the design of AWE systems with respect to these different
profit-based metrics will be relevant once we have understood the design with respect to LCoE.

The described system design framework, including the individual models, is implemented in MATLAB and is available open source through a GitHub repository (Joshi, 2024). The following section presents a case study showcasing the functionality of the framework.

## 3    Case study

The case study determines optimal system configurations for rated power of 100, 500, 1000, and 2000 kW. A design space is then explored to identify configurations that minimize the LCoE for each rated power. A reference scenario is defined in the next section, followed by a sensitivity analysis.

### 3.1    Reference scenario

Table 6 lists the fixed parameters defining the reference scenario. The input wind resource to the cycle power model is defined
by the combination of the wind speed at a fixed reference height $h_{\text{ref}}$ and the vertical wind profile shape at that location. The chosen wind conditions of $v_{\text{w,mean}} = 8.5 \text{ms}^{-1}$ and $\alpha_{\text{w}} = 0.2$ correspond to Class I wind turbine conditions as described in



International Electrotechnical Commission (IEC) (2019). The cut-in and rated wind speeds depend on the design variables, but the cut-out wind speed is assumed constant at 25 ms$^{-1}$ at the operational height. In later design stages, the cut-out wind speed will be determined using higher-fidelity engineering analyses. The wind speed limits are always with respect to the reference height $h_{\mathrm{ref}}$. Since the framework does not employ an aerodynamic model, the wing aerodynamic properties are assumed constant for all designs. The underlying assumption is that the same airfoil is used for all the kite sizes. To account for a stall-safety margin and the 3-D wing aerodynamic effects, an airfoil efficiency factor $\eta_{\mathrm{Cl}}$ is applied on the maximum airfoil lift coefficient $C_{\mathrm{l,max}}$ to impose an upper limit for the wing lift coefficient as

$$C_{\mathrm{L,max}} = \eta_{\mathrm{Cl}} C_{\mathrm{l,max}}. \tag{36}$$

A land surface area constraint is not applied since this would be more relevant for farm-level studies. Neglecting this constraint will allow us to understand the unconstrained potential of single systems. Similarly, the maximum operating height constraint is also not applied, as it is primarily driven by airspace regulations and is highly location-dependent. This is done by setting a relatively high upper limit of 1000m. On the other hand, a minimum operating height of 100 m is applied as a safety constraint. The maximum tether reeling speed $v_{\mathrm{drum,max}}$ and acceleration $a_{\mathrm{drum,max}}$ are a result of the limits driven by the drum dynamics.

**Table 6.** Fixed parameters describing the reference scenario.

| Parameter | Description | Value | Unit |
|---|---|---|---|
| $\alpha_{\mathrm{w}}$ | Wind shear coefficient | 0.2 | - |
| $h_{\mathrm{ref}}$ | Reference height | 100 | m |
| $v_{\mathrm{w,mean}}$ | Mean wind speed at ref. height | 8.5 | ms$^{-1}$ |
| $k$ | Weibull shape parameter | 2 | - |
| $r$ | Discount rate | 0.10 | - |
| $N_{\mathrm{y}}$ | Project lifetime | 25 | years |
| $C_{\mathrm{l,max}}$ | Max. airfoil lift coefficient | 2.5 | - |
| $\eta_{\mathrm{Cl}}$ | Airfoil efficiency factor | 0.80 | - |
| $C_{\mathrm{l,Cd,min}}$ | Lift coefficient at minimum drag coefficient | 0.65 | - |
| $C_{\mathrm{d,min}}$ | Minimum drag coefficient | 0.056 | - |
| $\rho_{\mathrm{t}}$ | Tether material density | 970 | kgm$^{-3}$ |
| $C_{\mathrm{d,t}}$ | Cross-sectional tether drag coefficient | 1.2 | - |
| $h_{\mathrm{min}}$ | Min. ground clearance | 100 | m |
| $h_{\mathrm{max}}$ | Max. operating height | 1000 | m |
| $v_{\mathrm{drum,max}}$ | Max. tether reeling speed | 20 | ms$^{-1}$ |
| $a_{\mathrm{drum,max}}$ | Max. tether reeling acceleration | 5 | ms$^{-2}$ |
| $N_{\mathrm{p,min}}$ | Minimum number of patterns per cycle | 1 | - |





The following section shows the design space exploration results for the rated power of 500 kW.

### 3.1.1 Design space exploration for a 500 kW system

Table 7 shows the design space explored using the framework described in Sect. 2.2 for a system rated power of 500 kW. The variables are as defined in Sect. 2.1.2, and the optimisation objective is the LCoE as described in Sect. 2.1.1. An even wider space was investigated, but only part of this space, around the optimal solution, is discussed in the following.

**Table 7.** Explored design space for 500kW rated power.

| Variable | Range [min, max] | Step size | Unit |
|---|---|---|---|
| $S$ | $[50, 70]$ | 10 | $\mathrm{m}^2$ |
| AR | $[10, 14]$ | 2 | - |
| $W_{\mathrm{l,max}}$ | $[2, 4]$ | 1 | $\mathrm{kNm}^{-2}$ |
| $\sigma_{\mathrm{t,max}}$ | $[0.3, 0.5]$ | 0.1 | GPa |
| $f_{\mathrm{crest}}$ | $[1.5, 2.5]$ | 0.5 | - |

Two variables are varied independently, and the results are illustrated as LCoE contour plots and representative power curves. The combinations of variables are chosen based on the degree of the coupling between the two variables. The other variables are kept constant during this process. For example, the wing area is varied with maximum wing loading since they are coupled through the tether force. Maximum wing loading is varied with maximum allowable tether stress since they are coupled through the tether diameter and characterise the tether. The wing aspect ratio is varied with wing area since they characterize the kite. The power crest factor is varied with the wing area since the wing area significantly influences the extractable power and the power crest factor limits this power by limiting the generated rated power. Overall, the wing area is a key parameter characterizing the size and power output of GG AWE systems.

Figure 7 and Fig. 8 show the computed LCoE and associated power curves, respectively, for varying wing area and maximum wing loading. The optimal wing area and maximum wing loading values resulting in the minimum LCoE of 142 €/MWh are 60 $\mathrm{m}^2$ and 3 $\mathrm{kNm}^{-2}$, respectively. Limiting the wing area and the wing loading to lower values limits the power extraction and increases the LCoE. Some configurations cannot reach the specified rated power target of 500 kW as seen in Fig. 8. These are the combinations with smaller wing areas and smaller maximum wing loading. Increasing the area and loading of the wing increases the kite mass, which in turn increases the losses due to gravitational effects, resulting in an increased LCoE. The system configurations with lower LCoEs are the combinations of smaller wing areas with higher wing loading or larger wing areas with lower wing loading, consequently resulting in the optimum value, as seen in Fig. 7.

Figure 9 and Fig. 10 show the computed LCoE and associated power curves, respectively, for varying maximum wing loading and tether stress. Maximum wing loading primarily affects the maximum power output, while maximum tether stress has a major influence on the tether lifetime. The combined effect of these two variables on the LCoE is highly non-linear. The



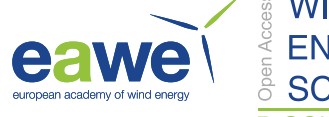

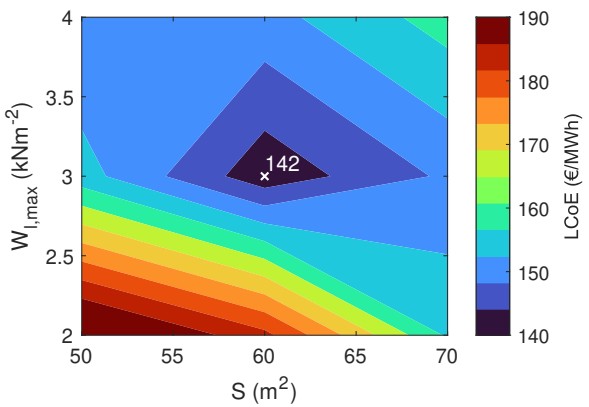

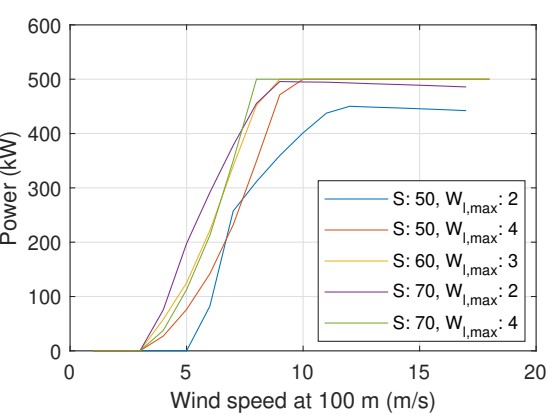

**Figure 7.** LCoE as a function of wing area $S$ and maximum wing loading $W_{l,max}$. The other design variables are held constant with the following values: aspect ratio $AR = 12$, maximum tether stress $\sigma_{t,max} = 0.4$ GPa, and power crest factor $f_{crest} = 2$.

**Figure 8.** Power curves of a few system configurations within the design space illustrated in Fig. 7. The configurations with smaller wing areas and smaller maximum wing loading cannot reach the rated power of 500kW.

two parameters also affect the tether diameter, thereby impacting the tether drag losses. The minimum LCoE of 142 €/MWh is reached for the combination of max wing loading of $3$ kNm$^{-2}$ and a maximum tether stress of $0.4$ GPa.

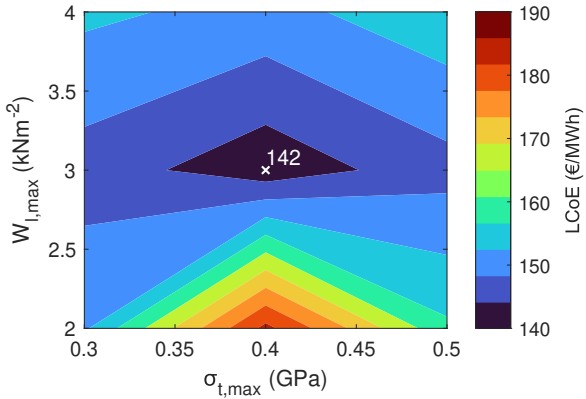

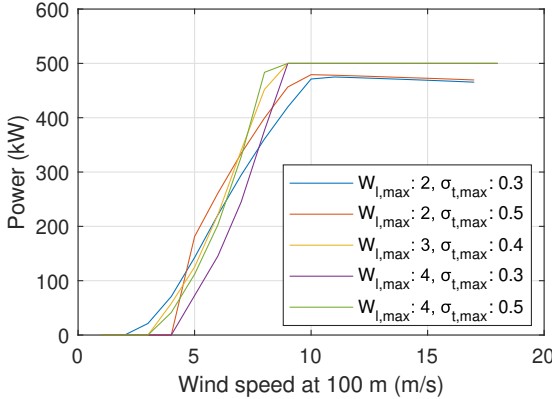

**Figure 9.** LCoE as a function of maximum wing loading $W_{l,max}$ and maximum tether stress $\sigma_{t,max}$. The other design variables are held constant with the following values: wing area $S = 60$ m$^2$, aspect ratio $AR = 12$, and power crest factor $f_{crest} = 2$.

**Figure 10.** Power curves of a few system configurations as illustrated in Fig. 9. The configurations with lower values of maximum wing loading cannot reach the target rated power of 500 kW.



Figure 11 and Fig. 12 show the computed LCoE and associated power curves, respectively, for varying wing area and aspect ratio. Compared to other variables, it is observed that the aspect ratio has a small influence on the LCoE. The figure indicates that the LCoEs computed with aspect ratios of 10 and 12 are very similar. Unlike the configurations from Figs. 8 and 10, all the combinations in this design space reach the target rated power of 500 kW. The variation of wing area has a major effect on the power curve, whereas, for a given wing area, the effect of variation of the aspect ratio is minimal. This is the reason for the clusters of power curves, as seen in Fig. 12. The rated power is achieved at relatively lower wind speeds with increasing wing area.

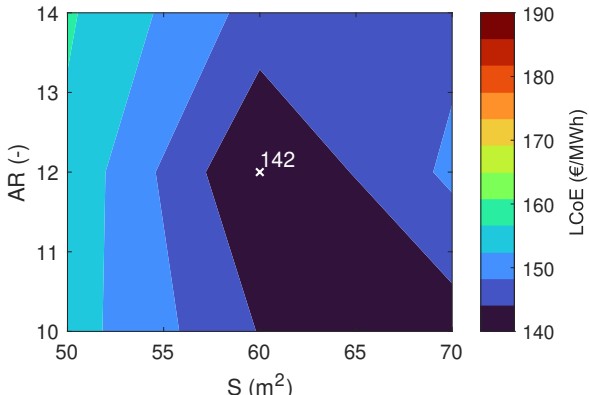

**Figure 11.** LCoE as a function of wing area and aspect ratio. The other design variables are held constant with the following values: maximum wing loading $W_{l,\max} = 3\text{kNm}^{-2}$, $\sigma_{t,\max} = 0.4\text{GPa}$, and power crest factor $f_{\text{crest}} = 2$.

**Figure 12.** Power curves of a few system configurations as illustrated in Fig. 11. The configurations with the same wing area are clustered together since the influence of the aspect ratio is relatively smaller than the influence of the wing area.

Figure 13 and Fig. 14 show the computed LCoE and associated power curves, respectively, for varying wing area and the power crest factor. Since the power crest factor limits the rated generator power, consequently limiting the maximum reel-out power, the power curves show that for smaller crest factors, the system cannot reach the rated power of 500 kW. A higher crest factor means larger drivetrains and, hence, higher costs. Since the rated power is capped at 500kW, a crest factor of two gives the minimum LCoE.

Based on the above results, the optimal system configuration produces the power curve depicted in Fig. 15. The values of the design variables are $S = 60\text{m}^2$, $AR = 12$, $W_{l,\max} = 3\text{kNm}^{-2}$, $\sigma_{t,\max} = 0.4\text{GPa}$, $f_{\text{crest}} = 2$. The cut-in, rated, and cut-out wind speeds are 6, 11, and $20\text{ms}^{-1}$ at the reference height of 100 m.

Figure 16 illustrates the shares of the subsystems in the total capital expenditure (CapEx), operational expenditure (OpEx) and the LCoE. The CapEx is dominated by the kite structure costs resulting directly from the kite mass and the ultracapacitor costs resulting from the power smoothing requirement. The OpEx is composed of the replacements of the ultracapacitors and the tether, along with the balance of system costs. These are again reflected in the LCoE split. Compared to horizontal axis wind turbines (HAWTs), one of the key characteristics is the ratio of total CapEx and the lifetime OpEx. In this case, the





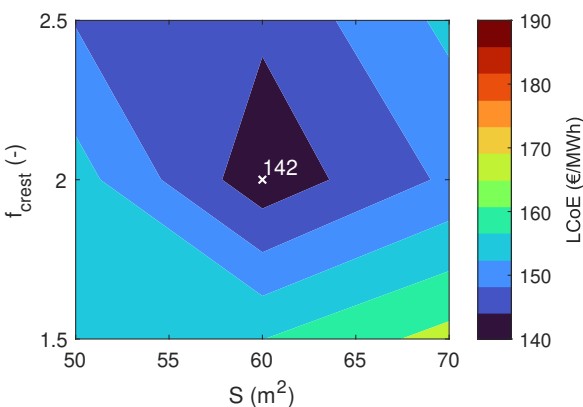

**Figure 13.** LCoE as a function of wing area and power crest factor. The other design variables are held constant with the following values: aspect ratio $AR = 12$, maximum wing loading $W_{l,max} = 3\text{kNm}^{-2}$, and $\sigma_{t,max} = 0.4\text{GPa}$.

**Figure 14.** Power curves of a few system configurations as illustrated in Fig. 13. The maximum reel-out power of configurations with a power crest factor $f_{crest} = 1.5$ is capped at 750 kW and hence they can only attain a rated power of 400 kW.

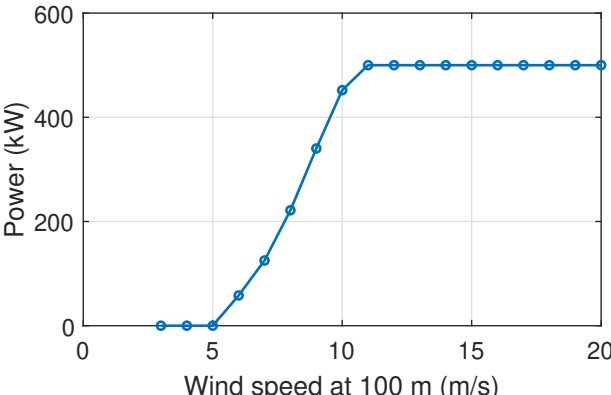

**Figure 15.** Power curve of the 500 kW system based on the optimal system design minimising the LCoE.

undiscounted OpEx, considering a lifetime of 25 years, is 2600 k€, which is larger than the CapEx. This indicates that the GG AWE systems do not have high upfront costs but more spreadout costs which can be an advantage in terms of financing, as compared to HAWTs.

A similar detailed analysis was performed for the rated power of 100, 1000, and 2000 kW. The following section jointly investigates all power ratings to derive insights into the scaling behaviour of fixed-wing GG AWE systems.





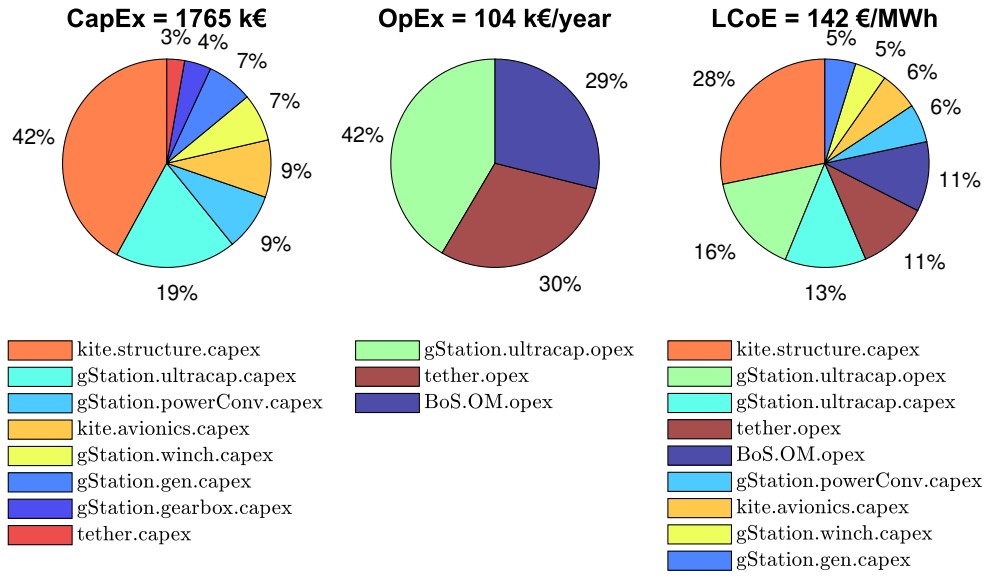

**Figure 16.** Share of subsystem component costs ($\geq 2\%$) within the capital expenditure (CapEx), operational expenditure (OpEx) and the LCoE. The terminology and the nomenclature used in the legend are described in Joshi and Trevisi (2024).

### 3.1.2 Scaling trends with 100 kW to 2000 kW systems

Table 8 lists the explored design space with the system design variables as defined in Sect. 2.1.2. The lower and upper limits of the design variables are based on the design space explored for the 500 kW system, available prototype data from companies, and engineering guesses. Our analysis showed that the optimum values always lie within the design space considered in this study for chosen system sizes.

**Table 8.** Explored design space for the rated power of 100, 1000, and 2000 kW.

| Variable | Range [min, max] | Step size | Unit |
|---|---|---|---|
| $S$ | $[10, 170]$ | 10 | m$^2$ |
| AR | $[8, 14]$ | 2 | - |
| $W_{l,\max}$ | $[1, 5]$ | 1 | kNm$^{-2}$ |
| $\sigma_{t,\max}$ | $[0.3, 0.5]$ | 0.1 | GPa |
| $f_{crest}$ | $[1.5, 2.5]$ | 0.5 | - |

Figure 17 and Fig. 18 show the computed LCoE, capacity factor (cf) and the corresponding power curves for the optimal system configurations. The computed LCoE is minimum for the 500 kW size, while the capacity factor monotonously decreases with increasing system size.




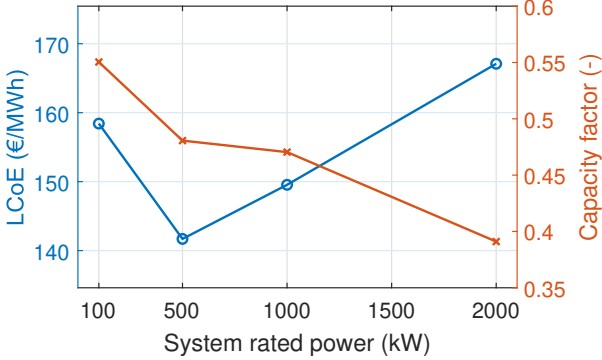

**Figure 17.** LCoE and capacity factor (cf) of the optimal system configurations for the four specified rated powers in the reference scenario.

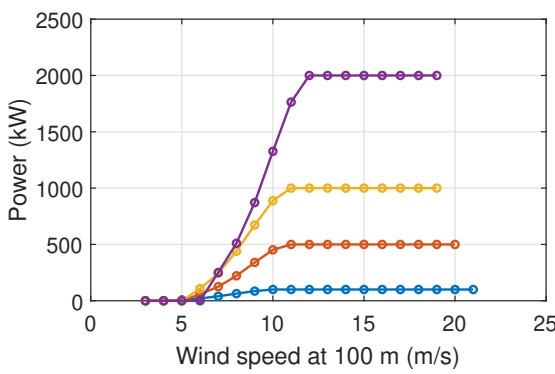

**Figure 18.** Power curves of the optimal system configurations for the four specified rated powers in the reference scenario.

Table 9 shows the values of the design variables and some key specifications describing the optimal configurations minimising the LCoE for the four rated powers. A linear relationship between the rated power and the optimal wing area does not exist. The kite area specific power for the 100 kW system is $5\,\mathrm{kWm^{-2}}$ and for the 2000 kW system is $12.5\,\mathrm{kWm^{-2}}$. These results indicate that the costs increase faster with size than the energy produced. The cut-in and the rated wind speeds for the optimal configurations increase with size and hence the decrease in the capacity factor as seen in Fig. 17.

**Table 9.** Optimised values of the design parameters and some key resulting specifications that minimise the LCoE for the four rated powers.

| $P_{\mathrm{rated}}$ (kW) | 100 | 500 | 1000 | 2000 |
|---|---|---|---|---|
| $S$ (m$^2$) | 20 | 60 | 110 | 160 |
| $AR$ $(-)$ | 10 | 12 | 10 | 10 |
| $W_{\mathrm{l,max}}$ (kNm$^{-2}$) | 2 | 3 | 3 | 4 |
| $\sigma_{\mathrm{t,max}}$ (GPa) | 0.4 | 0.4 | 0.4 | 0.4 |
| $f_{\mathrm{crest}}$ $(-)$ | 2 | 2 | 2 | 2 |
| $m_{\mathrm{k}}$ (kg) | 700 | 2792 | 5857 | 10663 |
| $d_{\mathrm{t}}$ (cm) | 1.13 | 2.39 | 3.24 | 4.51 |
| $v_{\mathrm{w,cut-in}}$ (ms$^{-1}$) | 5 | 6 | 6 | 7 |
| $v_{\mathrm{w,rated}}$ (ms$^{-1}$) | 10 | 11 | 11 | 12 |
| $v_{\mathrm{w,cut-out}}$ (ms$^{-1}$) | 21 | 20 | 19 | 19 |
| CapEx (k€/yr) | 495 | 1765 | 3656 | 6864 |
| OpEx (k€/yr) | 21 | 104 | 213 | 388 |





### 3.1.3 Power harvesting factor, specific power, and coefficient of power trends

A commonly used non-dimensional metric in the literature to quantify the performance of AWE systems is the power harvesting factor $\zeta$ (Diehl, 2013). It is defined as the ratio of the extracted power to the kinetic energy flux through a cross-sectional area equal to the wing area,

$$\zeta = \frac{P}{\frac{1}{2}\rho S v_{\mathrm{w}}^3}. \tag{37}$$

Figure 19 depicts the computed values for the optimal system configurations. The trend shows that for LCoE-optimised systems, the extractable power per unit wing area shows diminishing marginal gain with increasing wing area.

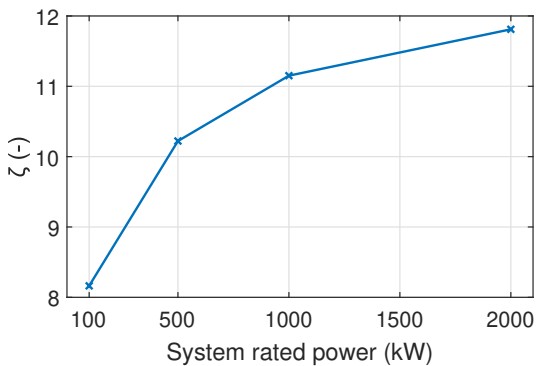

**Figure 19.** Power harvesting factors of the optimal system configurations.

The specific power of horizontal axis wind turbines (HAWTs) is defined as the ratio of the rated turbine power to the rotor-swept area. The specific power of HAWTs designed for different markets and wind speed classes are within the range of 200-400 $\mathrm{W m^{-2}}$ (Mehta et al., 2024a). The turbines at the lower end of this range are designed for lower wind speed sites to maximise the energy capture. Since the swept area of AWE systems generally varies with the operation and control strategies, a definition based on the swept area is not practical. Therefore, we define the specific power for AWE systems using the kite wing area instead of the swept area as

$$SP_{\mathrm{S}} = \frac{P_{\mathrm{rated}}}{S}. \tag{38}$$

Figure 20 illustrates this specific power and maximum wing loading for the LCoE-optimised system configurations. Both parameters increase with rated power. In contrast to this, the LCoE-optimised HAWTs have a constant specific power in the range of 200-400 $\mathrm{W m^{-}2}$ (Mehta et al., 2024a), irrespective of rated power. This shows the difference in scaling behaviours of GG AWE systems and HAWTs. Together, both of these trends can indicate the choice of wing area and tether combination for any power rating that minimises LCoE. The results listed in Table 9 are characterised by a constant optimal power crest factor of two. Hence, in addition to the wing area and tether combination, the drivetrain size that minimises LCoE will be two times that of the targeted rated power.

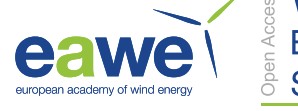


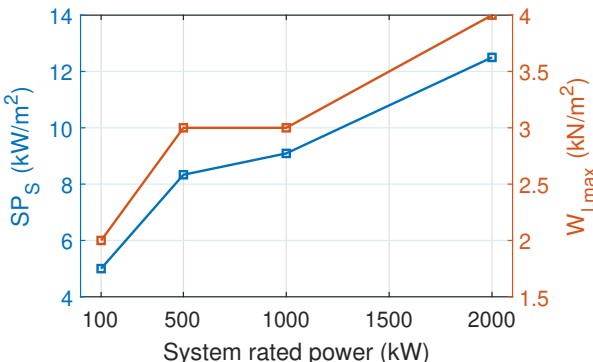

**Figure 20.** Specific power using the kite wing area on the left axis and maximum wing loading on the right axis. The plateau in the maximum wing loading trend is due to the step size used in the design space.

Trevisi et al. (2023) used a different reference area to define a power coefficient and specific power, given as

$$A_{\mathrm{ref}} = \pi b^2. \tag{39}$$

Geometrically, this represents the area of a circle using the wing span of the kite as the radius. This is analogous to calculating the swept area of wind turbines using the blade length as the radius. The resulting coefficient of power is defined as

$$C_{\mathrm{p,Aref}} = \frac{P_{\mathrm{rated}}}{\frac{1}{2}\rho A_{\mathrm{ref}} v_{\mathrm{w,rated}}^3}. \tag{40}$$

Figure 21 depicts the specific power and power coefficient using the above definition of the reference area. Both parameters are increasing with kite size similar to Fig. 20. The order of magnitude corresponds to that commonly observed for HAWTs, though the definition of the parameters is different. Though the power output per reference area increases with increasing rated power, it shows that the LCoE-optimised system is not an energy-yield-optimised system.

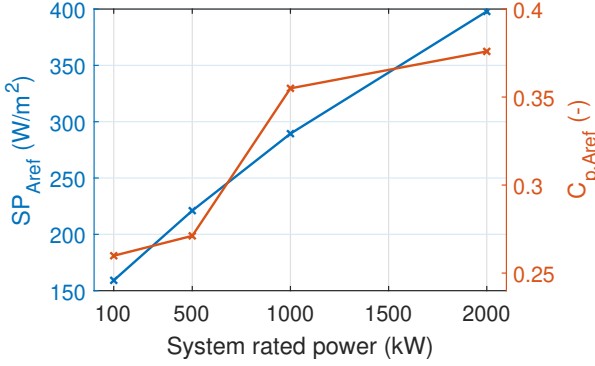

**Figure 21.** Specific power and power coefficient using the reference area definition as proposed by Trevisi et al. (2023).





## 3.2 Scenario sensitivity

To understand the sensitivity of the presented solutions, we investigated the deviations from the reference scenario as described in Table 10. These deviations represent more extreme scenarios in technological improvements, environmental conditions, and market characteristics. The sensitivity with respect to kite mass (scenario 1) and ultracapacitor costs (scenario 2) was considered since these components are dominating the LCoE. Moreover, the grid operator may allow for electricity to be taken from the grid during reel-in, which will take away the storage costs required for power smoothing. The sensitivity to the discount rate (scenario 3) is of interest because geopolitical phenomena such as recession or inflation can affect the interest rates. Scenario 4 and 5 capture extreme variations of the wind conditions.

**Table 10.** Scenarios defined for sensitivity analysis in comparison with the reference scenario.

| No. | Scenario | Assumptions against the Reference |
|-----|----------|-----------------------------------|
| 1 | Reduced $m_k$ by 50% | Steep technological advancements reducing the kite mass by 50 % |
| 2 | No storage | No power smoothing requirement from grid |
| 3 | Increased $r$ to 15% | Higher discount rates due to uncertainties |
| 4 | $\alpha_w = 0$ | Environmental conditions with no wind shear ($\alpha_w$ is the wind shear coefficient) |
| 5 | $\alpha_w = 0, v_{w,mean} = 10\text{ms}^{-1}$ | No wind shear but high mean wind speed, representing wind turbine Class I conditions |

### 3.2.1 Scaling trends

Figure 22 depicts the LCoE for the optimal system configurations in all considered scenarios. It can be seen that the optimal

system size with the minimum LCoE is 500 kW in all scenarios. As expected, the reduced kite mass and the removal of storage costs further reduce the LCoE values for all systems. These curves are flatter than the other scenarios, indicating that the optimal system size will likely shift towards larger power ratings when accounting for technological improvements.

Compared to Table 1, which lists the LCoE values for AWE systems reported in the public domain, the LCoE values found within this study lie on the upper end of the spectrum. Moreover, Gambier et al. (2014, 2017) had a similar finding to our

study about soft-wing systems that the LCoE had a minimum at 200 kW rated power and increased with further upscaling. IRENA (2023) reported the global averages of LCoEs for different renewable energy technologies in 2023. The onshore wind was around 33 €/kWh, utility-scale solar PV was around 44 €/kWh, and offshore wind was around 75 €/kWh. All these technologies have experienced steep reductions in LCoE over the past half-century due to technological advancements and maturity. A similar trend could likely be predicted for AWE systems, which will further reduce the LCoE values in the next

decade.



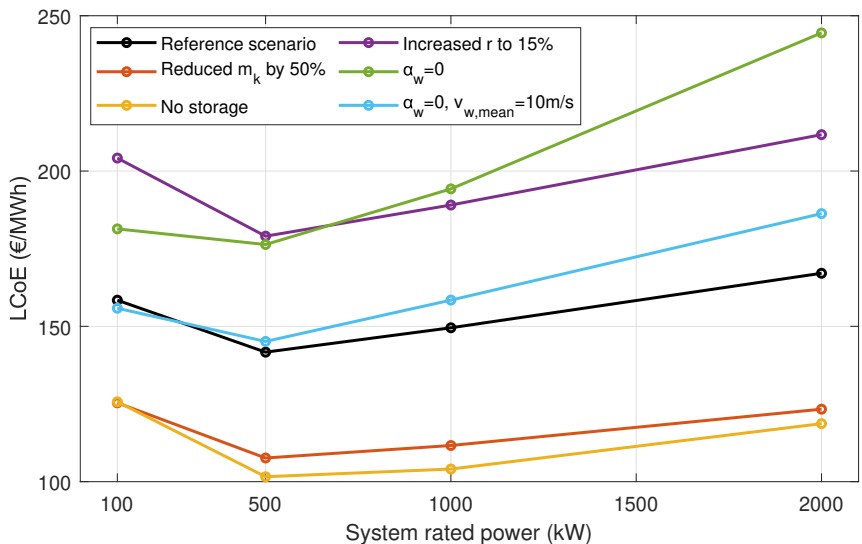

**Figure 22.** LCoE vales for the four rated powers evaluated for the considered scenarios in comparison to the reference.

### 3.2.2 Optimal system configurations

Table 11 lists the optimal values for the design parameters and some of the key resulting system specifications that minimise the LCoE of the 500 kW system. The computed values vary slightly across all scenarios based on their effect. For example, the optimal wing area in scenario 1, considering a reduced kite mass, is larger since the penalty due to the gravitational force is lower than in the reference scenario. This also allows for a larger tether diameter by increasing the maximum tether stress value, thereby reducing the tether replacement costs.

### 3.3 Discussion

Unlike HAWTs, fixed-wing GG AWE systems do not show distinct upscaling benefits when scaling up to megawatts. The unfavourable scaling of kite mass drives this outcome, as the kite has to use part of its aerodynamic force to compensate for gravity, which is increasingly penalising with size. Since this study is focused on single systems, farm-level aspects which can influence scaling are not reflected in the results. Having fewer systems in a single farm reduces the installation, operation and maintenance costs and hence would motivate larger individual systems. Also, area constraints will likely drive the solution towards larger systems to reduce the overall farm-LCoE. As a result of these effects not being considered in this study, the optimum system size could potentially increase due to technological improvements in materials and manufacturing methods decreasing the kite mass. The presented framework can be coupled with approaches that evaluate an entire energy system to analyse larger-scale effects. Malz et al. (2022) looked at the value of AWE farms to the electricity system based on a metric known as the marginal system value (MSV). This metric quantifies the additional value that one extra unit of electricity





**Table 11.** Optimised values of the design parameters and some key resulting system specifications that minimise the LCoE of the 500 kW system in all scenarios.

| Scenario | Reference | 1 | 2 | 3 | 4 | 5 |
|---|---|---|---|---|---|---|
| $S$ (m$^2$) | 60 | 70 | 50 | 60 | 60 | 60 |
| $AR$ $(-)$ | 12 | 10 | 10 | 12 | 10 | 8 |
| $W_{\mathrm{l,max}}$ (kNm$^{-2}$) | 3 | 3 | 3 | 3 | 3 | 3 |
| $\sigma_{\mathrm{t,max}}$ (GPa) | 0.4 | 0.3 | 0.4 | 0.4 | 0.4 | 0.4 |
| $f_{\mathrm{crest}}$ $(-)$ | 2 | 2 | 2 | 2 | 2 | 2 |
| $m_{\mathrm{k}}$ (kg) | 2729 | 1634 | 2164 | 2792 | 2700 | 2682 |
| $d_{\mathrm{t}}$ (cm) | 2.39 | 2.99 | 2.19 | 2.39 | 2.39 | 2.39 |
| $v_{\mathrm{w,cut-in}}$ (ms$^{-1}$) | 6 | 5 | 6 | 6 | 7 | 7 |
| $v_{\mathrm{w,rated}}$ (ms$^{-1}$) | 11 | 10 | 12 | 11 | 13 | 14 |
| $v_{\mathrm{w,cut-out}}$ (ms$^{-1}$) | 20 | 19 | 19 | 20 | 25 | 25 |
| CapEx (k€) | 1765 | 1490 | 1249 | 1765 | 1673 | 1639 |
| OpEx (k€/yr) | 104 | 94 | 51 | 104 | 95 | 111 |

generated by the AWE system brings to the overall energy system. In their analysis, they included vertical wind profiles and optimised the flight trajectories of AWE systems to maximum average power. Their overarching conclusion was that AWE systems and wind turbines are interchangeable technologies since they have similar power production profiles. Malz et al. (2022) found that small AWE systems generally have more full-load hours than large systems, which aligns with our finding of decreasing capacity factors with increasing size. A key difference in their approach was that MSV is a cost-independent metric which tries to quantify the added benefit of AWE systems in terms of energy production. Vos et al. (2024) conducted a study on the integration of AWE at the European energy system level. In their analysis, they used the versions of the power and cost models used in our present work, which were then still in the development phase. They also concluded that the AWE systems perform similarly to the wind turbines in offshore scenarios, and the competitiveness is heavily dependent on the costs. However, Vos et al. (2024) found that AWE systems have an advantage onshore due to better wind resource availability at higher altitudes than the average hub heights of wind turbines. The conclusions of these earlier studies align with the findings of the present work, but it will be beneficial to perform such studies again using the models and the presented system designs.

## 4 Conclusions

In contrast to HAWTs, AWE systems use tethered flying devices to harvest wind energy. The fundamentally different working principles and resulting system designs lead to a different scaling behaviour of the technology. The rotor-nacelle assembly of HAWTs is positioned in the flow using a tower. In AWE systems, the kite has to use part of its aerodynamic force to counter the gravitational loading of the airborne system parts. An MDAO framework was developed to understand the scaling behaviour





of fixed-wing GG AWE systems using LCoE as the design objective to conduct a holistic system performance assessment. System design parameters such as the wing area, aspect ratio, maximum wing loading, maximum tether stress and the power crest factor were chosen as independent variables to systematically explore the design space. These parameters were optimised for the system sizes of 100, 500, 1000 and 2000 kW.

     The minimum LCoE was found for the 500 kW system, and the extractable power per unit wing area shows diminishing

marginal gain with increasing wing area. This shows that there is no distinct benefit in upscaling the systems to multiple megawatts in terms of LCoE. This outcome is due to the penalising effect of the kite's weight on energy production and costs. Increasing rated power demands a larger kite, and since the mass increases rapidly with size, this has a negative effect since part of the aerodynamic force is used to counter the gravitational force. As a result, there is an increase in the cut-in wind speed, followed by an increase in the rated wind speed. Therefore, similarly to conventional wind turbines, we also see a decrease

in capacity factor with increasing rated power. The primary cost-driving components for fixed-wing GG AWE systems are the kite mass, storage replacements and tether replacements. Unlike conventional wind turbines, the total lifetime operational costs of AWE systems are equal to or even exceed the initial investment costs. This distribution of expenses over the project's lifetime reduces upfront investments for project financing, which will have significant implications, particularly in markets where securing substantial initial investments is challenging. Sensitivity analyses were performed with scenarios representing

extreme environmental conditions, financial assumptions and technological improvements. These results show the same scaling trend indicating sufficient robustness of the conclusions made in this work.

     We suggest that academic efforts such as defining reference models and developing higher-fidelity tools, together with industrial efforts to develop commercial products, should target the system sizes in the 500 kW-range. The results show the importance of focusing research efforts on kite design, primarily to reduce mass by investigating innovative materials and

manufacturing techniques. Additionally, we recommend research to improve tether design and system operation to increase fatigue life and minimise replacements. Factors that can drive the optimum to larger systems could be farm-level effects since having fewer systems generally reduces the overall installation and operation costs. Though the analysis presented in this paper was focused on fixed-wing GG systems, the key conclusions about the scaling behaviour will most likely hold for other concepts as well. Moreover, using models tuned for other concepts, the described methodology can obtain specific insights.

Such system-level insights are important to guide the research and development of AWE technology.

*Code availability.* The model is implemented in MATLAB and is available on GitHub from https://github.com/awegroup/AWE-SE. It contains pre-defined input files which can be used to run the model and reproduce the results presented in the paper.

## Appendix: Nomenclature

### Greek symbols

$\alpha$         Wind shear coefficient



| | |
|---|---|
| $\beta$ | Elevation angle |
| $\eta$ | Efficiency |
| $\gamma$ | Cone opening angle |
| $\rho$ | Material density |
| $\sigma$ | Material strength |
| $\zeta$ | Power harvesting factor |

**Latin symbols**

| | |
|---|---|
| $A$ | Area |
| $a$ | Acceleration |
| $C$ | Capital cost |
| $C_{\mathrm{D}}$ | Drag coefficient |
| $C_{\mathrm{L}}$ | Lift coefficient |
| $D$ | Drag |
| $d$ | Diameter |
| $E$ | Energy |
| $e$ | Wing planform efficiency factor |
| $F$ | Force |
| $f$ | Factor |
| $h$ | Height |
| $L$ | Lift |
| $l$ | Length |
| $m$ | Mass |
| $N$ | Number |
| $O$ | Operation and maintenance cost per year |



| 685 | $P$ | Power |
| | $p$ | Price per unit |
| | $R$ | Radius |
| | $r$ | Discount rate |
| | $S$ | Wing area |
| 690 | $t$ | Time |
| | $v$ | Velocity |
| | $z$ | Z-axis co-ordinate |
| | AR | Aspect ratio |

**Subscripts**

| 695 | avg | Average |
| | BoS | Balance of system |
| | decomm | Decommissioning |
| | DT | Drivetrain |
| | e | Electrical |
| 700 | eff | Effective |
| | found | Foundation |
| | gb | Gearbox |
| | gen | Generator |
| | i | Reel-in |
| 705 | install | Installation and commissioning |
| | k | Kite |
| | l | Loading |
| | m | Mechanical |



| | | |
|---|---|---|
| | mat | Material |
| 710 | max | Maximum |
| | min | Minimum |
| | o | Reel-out |
| | p | Pattern |
| | pc | Power converters |
| 715 | ref | Reference |
| | sto | Storage |
| | str | Structure |
| | t | Tether |
| | uc | Ultracapacitor |
| 720 | w | Wind |
| | y | Instantaneous year |

*Author contributions.* Conceptualisation, R.J., D.v.T. and R.S.; methodology, R.J. and D.v.T.; software, R.J.; validation, R.J., D.v.T. and R.S.; writing—original draft preparation, R.J.; writing—review and editing, D.v.T. and R.S.; supervision, D.v.T. and R.S.; funding acquisition, R.S.

*Competing interests.* At least one of the (co-)authors is a member of the editorial board of Wind Energy Science.

*Acknowledgements.* The development of the associated code repository was supported by the Digital Competence Centre, Delft University of Technology.

*Financial support.* This research has been supported by the Nederlandse Organisatie voor Wetenschappelijk Onderzoek (grant no. 17628)





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
