# Peer review of "System design and scaling trends in airborne wind energy demonstrated for a ground-generation concept"

_Wind Energy Science, 2024_

## Author Comment (AC1)

**Round 1**

**Authors' response to Reviewer 1 comments**

Rishikesh Joshi

February 7, 2025

We appreciate your feedback and comments on our manuscript. Following are our responses.

The provided comments are in the standard black font, and our responses to the comments are in blue. The associated changes are in the revised manuscript submitted with this document.

**Provided comments**

1. There seems to be no variance between the mass dependence of horizontal take-off and landing (HTOL) kites, ground-generation kites (ground gen) and air-generation kites (air-gen).

   a. HTOL kites would expect higher structural mass to handle landing loads, but probably better than VTOL configuration

   b. VTOL lift-based kites would expect added mass due to VTOL weight as mentioned in the article

   c. VTOL kites with air-gen would expect added mass of VTOL weight, as well as higher drag of tether due to having a conductive tether (thicker diameter and more mass).

      i. Perhaps add a comment on system mass depending on configuration and how this is (not?) accounted for in the analysis

   Correct, an explanatory text is now included in Section 2.3: Kite mass model.

2. The article uses a lot of colors. When printing the article for reading to do a proper review, it was difficult to discern differences between the meanings, and I had to use digital version for assessing the figures. Consider to add dashed lines for it to be readable in black and white.
   Figure 5 is changed to have different lines. Figures 7, 9, 11, and 13 are contour maps requiring colour differentiation; they are not changed. Figures 8, 10, 12, and 14 have curves in close proximity, and the combinations of dotted lines will eventually overlap, causing the same problem; hence, we decided to keep the colours for these figures. Figures 20, 21 and 22 are changed to include different markers.

3. Page 2 line 39: Recommend to use lift instead of thrust for power production. It is typically the lift of the kite that produces pulling force, not the thrust.
   We used the word 'thrust' since, in that section, we are making an analogy with horizontal axis wind turbines (HAWTs). But we agree that it can be confusing and hence now removed.

4. Looking quickly through the code base at github it is not clear how the tether drag is accounted for. I see that you mention it on page 10, where it is considered as a lumped at the kite, but it remains a question if this is correct, or representative enough. At the core, there are two main factors that limit the cut-in speeds: kite mass and tether drag where the latter tends to dominate. This means that the lumped model might not capture the tether drag effect to a sufficient degree.
   We have now cleaned the GitHub repository. The tether drag equation can be found in AWE-SE » AWE-Power » src » computePower.m » Lines 65-67. Indeed, the lumped tether drag approach is not highly accurate, but such an approach is widely used in AWE literature in steady or quasi-steady models. It has been shown to be representative enough in design studies by (Houska and Diehl, 2006; Argatove et al., 2009, Fechner and Schmehl, 2013; Joshi et al., 2024)

5. In terms of figure 22, and scaling of airborne wind, I would say that it is too early to say anything about scaling beyond 1MW, but that this is indicative based on our current understanding (and of course the square cube law in terms of mass). The output of this figure is also closely tied to the assumptions in the article.

   We agree that it is too early to state with high confidence about scaling beyond 1 MW. Hence, we have discussed our assumptions throughout the paper, and the limitations in the Discussion and Conclusions section. We therefore focus more on the trends and not the absolute numbers, as also stated in the Introduction.

---

## Author Comment (AC2)

**Round 1**

**Authors' response to Reviewer 2 comments**

**Rishikesh Joshi**

**February 7, 2025**

We appreciate your feedback and comments on our manuscript. Following are our responses.

The provided comments are in the standard black font, and our responses to the comments are in blue. The associated changes are in the revised manuscript submitted with this document.

**Provided comments**

1. Please define consistently and clearly throughout the paper which are independent and dependent variables.
   We have checked and corrected for instances where this was unclear. We have also added an additional table (Table 3 in the updated version) listing the operational design variables which are dependent on the system design variables. The last paragraph of Section 2.2 has additional text that explains the workflow better.

2. A variable dependency network diagram could be useful prior to the system design framework under 2.2
   The extended design structure matrix (XDSM) in Figure 3 shows the dependency through colour coding of design variables, coupling variables and the analysis blocks.

3. Lines 129 to 131 appear convoluted and not fully consistent. Please define what remains constant.
   We have modified this text for clarity.

4. In Line 140 and later one could change from "Developers" to "Developments" or "Technology Developments" to move from entity to process
   We decided to not change this since we do want to address the entities who would design the systems.

5. 144 Please correct "area of operation A_oper is the ground area density". Either area or areas density. If area density, the please define the quantity the area relates to. "Area per what"
   'Density' was a typo and is now removed from the text.

6. 622 and several times before: Please, always specify which cost you refer to OpEx or CapEx or even more detailed.
   Modified the text to include specifics.

7. 626 to 629. The repetition of the working principle in the conclusions is not required.
   Removed.

---

## Author Comment (AC3)

**Round 1**

**Authors' response to Reviewer 3 comments**

Rishikesh Joshi

February 7, 2025

We appreciate your feedback and comments on our manuscript. Following are our responses.

The provided comments are in the standard black font, and our responses to the comments are in blue. The associated changes are in the revised manuscript submitted with this document.

**Provided comments**

1. I suggest deleting the first sentence of the abstract. This paper doesn't want to venture into the can of worms of the connection between power rating and LCOE and whether HAWTs will continue to grow in rotor diameter and power rating or not. This is an AWE paper and the opening sentence should focus on AWE not HAWTs.
   Done.

2. After the first sentence of the introduction, it would be appropriate to explain why wind turbine sizes have grown. The increased power ratings and diameters have been associated with sharp decreases in the cost of energy for a given project size, such that wind is now one of the cheapest modes of generation. This context is otherwise missing.
   Done.

3. Page 2, line 36, suggest new paragraph before, "The rotor nacelle assembly...
   Done.

4. Page 3, line 50- I do not understand why onboard storage is required for the AWE system. Power electronics coupled to the generator suitably "smooth out" the grid signal for wind turbines. Why does that approach not work here? The addition of storage (ultracapacitors in this work) certainly adds a steep cost penalty to the design.
   The power electronics should be able to smoothen out the power within a certain range of variation, for example, slight variation in incoming wind speeds around mean wind speed. But in the case of ground-generation AWE systems, since the kite has to be reeled back in at the end of every cycle, the power goes up and down at least 100% of the cycle average value. This extreme variation requires an intermediate storage buffer. Please see Figure 6. This is also explained in detail in the referenced paper: Joshi et al., 2022.

5. This paper leans heavily on the prior work in "Power curve modeling and scaling of fixed-wind ground-generation airborne wind energy systems" by 2/3 of the same authors. I imagine there was some discussion at one point as to whether the two papers should be combined into one or kept separate. The first paper also delves into tradespace exploration for some of the same design variables and their link to power performance. I understand that this work focuses on LCOE and overall system sizing, but the similarities between the two papers should be addressed more explicitly than is currently done. Otherwise, it is unclear to the reader what the novel contribution to the literature is in this paper. I would suggest a clearly worded paragraph near the end of the Introduction that says something like, "Prior work by the authors presented the model and the power performance tradeoffs [cite]. This paper builds on the prior work and makes the novel contribution of..."
   Done.

6. I find the definition of design variables and constraints in sections 2.1.2 and 2.1.3 confusing. The design variables include maximum wing loading and maximum tether stress. These material/structural limits are more commonly applied as constraints in other MDO papers. I would have expected more tangible parameters such as tether diameter or tether/wing material as the DVs. The discussion of max wing loading and tether stress around Line 127 also uses

phrasing that makes them sound like constraints. Furthering my confusion is the list of optimized variables on Line 185 (reel-out stroke length, wind lift coefficient, kite speed, pattern radius, elevation angle, cone angle) that are not included in Table 2, yet are optimized as though they are DVs. My guess from the XDSM diagram is that there are different nestings of sizing optimizations occurring. For the purposes of discussion, better to mention all DVs in Section 2.1.2 and then also tag which modules/nesting level they are associated with.

We have modified the text describing the tether. For a given tether material, the tether diameter is directly coupled to the maximum force it can withstand. Since wing loading is the tether force normalised with wing area, essentially it is the tether diameter but normalised with wing area. This parameter drives the maximum extractable power by the system. Maximum tether stress is chosen as the second design variable characterising the tether since it directly affects the fatigue lifetime of the tether. For a given tether material, this parameter is also essentially the tether diameter but drives the fatigue lifetime instead of extractable power. The selection of tether diameter is hence captured by these two variables since they drive two different aspects. We have now added a table (Table 3 in the updated manuscript) describing the operational design variables. The explanation about the nested workflow is added at the end of Section 2.2.

7. Section 2.6.2 mentions a hollow-core fiber tether. Is there a copper wire/cable for communication, control, SCADA-type parameters coming to-and-from the kite?

For fly-generation systems, the hollow core is used to insert the conduction and communication cables. However, the hollow core, in principle, is the result of the manufacturing process. After loading, the tether diameter slightly shrinks through the core to give the worked-in diameter. Moreover, even in ground-generation systems, the core indeed could be used for communication cables.

8. I liked the inclusion of fatigue estimation for the tether! That is often tough to do in these conceptual design levels of fidelity.

Yes, the inclusion of tether fatigue is one of the novel aspects of this paper.

9. I believe there is some confusion with regards to BOS costs and the categorization of CapEx and OpEx. I am also suspicious that the cost comparison between AWE and HAWTs is not apples-to-apples. I therefore am not sold on the summary conclusion that AWE spreads out its costs over the lifetime better than HAWTs. I encourage the authors to review the annual Cost of Wind report from NREL (latest edition: https://www.nrel.gov/docs/fy25osti/91775.pdf) as that might clear some points up. The reason for my confusion and suspicion is:

- Line 407 says that BOS consists of "site preparation, ..., operation maintenance and decomissioning". This has me confused because BOS is considered an upfront CapEx expenditure, yet operation and maintenance is included? If so, then what constitutes OpEx?

  BoS has both, a CapEx component and an OpEx component. The site preparation, foundation, installation and decommissioning are the CapEx components of BoS. The OpEx component, as explained in Equation (34), includes all the yearly costs, for example, the lease of the land used and the insurance costs against potential risks and liabilities associated with their deployment and operation.

- Figure 16 shows CapEx to be purely the kite system cost and BOS to be included in the OpEx category. This is a different convention than I am used to.

  In Figure 16, the CapEx piechart not only shows the kite, but also the ground station and the tether. It is important to note that for readability, as mentioned in the caption, only the cost shares above 2% are the ones that are shown in the figure. In the OpEx piechart, BoS shows up because the OpEx component of BoS is greater than 2%.

- The authors need to cite a reference for wind energy costs to support their claim that AWE spreads out the costs over the lifetime more evenly. Somewhere between Lines 525-530 would be appropriate. The units in Figure 16 are not the same as the Cost of Wind reports, in addition to the difference in categorization of the costs, so I am not convinced by the claim but unable to do the mental math to prove the authors case.

  We have now used the Cost of Wind Energy report you mentioned in the earlier comment to provide numbers for this comparison. The text is now modified.

- In Line 605, the authors state that only a single kite was considered and no farm-level sizing, costs, or performance effects are accounted for. This is a bit surprising because this is a key driver for turbine upscaling. Fewer turbine positions for a given plant rating drives down the BOS costs and leads to lower LCOE. To compare AWE LCOE trends in the results here vs HAWCT LCOE trends in the marketplace without ensuring that the BOS cost trends are consistent makes me suspicious. I would suggest the authors provide a back-of-the-envelope calculation for HAWTs in a table here to ensure consistency in the numbers and assumptions. Finally, unless I missed it earlier, this assumption of looking at single turbine costs only should also be stated earlier.

  Yes, we understand that farm-level aspects will further influence AWE's scaling trends. However, since there

is no other study in the literature yet which explores the system design and scaling trends of AWE, we decided to focus on a single system as the first step. Farm-level aspects will require the inclusion of wake models and cabling models, which will heavily influence the ground area density of AWE farms. We do not have robust models to incorporate all the effects together. Hence, we decided to ignore the farm-level aspects in this study. We have now included this scope also in the Introduction. Making BoS trends consistent with HAWTs will not be an easy task since very little information is available regarding the BoS of AWE systems. We agree that the BoS cost modelling is one of the weakest parts of our cost model, but the key finding from our study is that the operational costs for AWE are high due to the tether and storage replacements. This is something that is non-existent for HAWTs. Moreover, we do not want to make explicit comparisons with HAWTs since we understand that it would not be fair unless we include all aspects together. We mention in the text of Section 3.2.1 and Section 3.3 that including farm-level effects might drive the optimum to larger power ratings.

10. Discussion in the paragraph at Line 540 might be easier to understand by mentioning that the square-cubed law is likely at play.
    Done.

11. Table 9 is labeled as "optimized", but this is *not* the output of an optimizer, correct? This is the "best performer" of the parametric study in Table 8? If so, I would remove the word "optimized" from the caption and the discussion in paragraph at Line 540. Furthermore, in that paragraph, only the kite area is discussed for the trends observed. What about discussion for why the other parameters change or stay constant?
    The table caption and the text are now modified to state clearly that the values which are stated as optimums are a result of an exhaustive parametric sweep within the defined design space. The word 'optimal' is used in the text in the sense of this parametric sweep. The discussion about parameters other than wing area is now added.

12. Why wasn't full system optimization used to generate Table 9 instead of just a parametric study with DV steps that some might argue are too coarse? I understand that it can be arduous to show consistent trends in a family of optimized designs as it requires lots of restarts at different initial conditions, etc, to ensure a robust and consistent output. However, without that I feel like the authors are not taking full advantage of the model capability they have presented and the insight into the design drivers and physics isn't as clear as it could be.
    We did perform such full-scale optimisations which gave us results which varied within only 3-4% with respect to the variables as well as the objective value. The time required to generate such results was significant, and the results did not change the observed trends. Hence, we instead decided to do a design space exploration with the step size based on a few full-scale optimisation iterations. Hence, the step size is not too coarse since higher resolution did not vary the trend and conclusions. On the other hand, the nested optimisation of the operational design variables is a full optimisation and not a parametric sweep.

    - Given the parametric approach taken in Table 9, I am also suspicious about Table 11, given the nice round numbers. Was that a true optimization or just the best performer is a parametric sweep?
      It is also the result of an exhaustive parametric sweep within the defined design space. The table caption is now modified for clarity.

13. The discussion in 3.2.1 around the capstone plot in Figure 22 could be stronger. Why is 500kW a robust optimal size for this AWE architecture across all of the scenarios? Even when mass is 50% less and wing area increases, the optimal rating stays the same- why? I believe the explanation that the kite has to use part of its aero lift to keep aloft, instead of using 100% for power generation, but does that explanation hold if a mass reduction by 50% doesn't have an impact on the design? What if the mass reduction were greater, or even in the 90-99% range? Is it because the steps in rated power are too coarse? What other hypothetical scenario would cause this optimal size to shift?
    The text in this section associating the figure is now extended. In the kite mass reduction scenario, the optimum does not shift since the mass reduction is applied for all power ratings. As also explained in the text, the curve does get flatter as compared to other scenarios, hinting that it is possible that with further aggressive mass reduction, the optimum could shift towards larger systems. As seen from the flatter curves, even if we increase the step size, the optimum power rating might be between 500-1000 KW, but this will have a small influence on the LCoE. Reducing the step size just for this scenario will likely not change the observed trends and conclusions. Another aspect which could increase the optimum power rating is the consideration of a farm setting. This is now also discussed in the text.

14. In Table 11, using the Scenario number to label the columns is confusing and requires lots of page flipping to understand what is going on. Some text-based shorthand to describe each column would help.
    Done.

15. If Table 11 is from a parametric search, please do not use the word Optimized in the caption (same comment as above).
Modified same as above.